# Updates and evaluation of NOAA's online-coupled air quality model version 7 (AQMv7) within the Unified Forecast System

Wei Li[1,2], Beiming Tang[1,2], Patrick C. Campbell[1,2], Youhua Tang[1,2], Barry Baker[1], Zachary Moon[1,3], Daniel Tong[1,2], Jianping Huang[4], Kai Wang[4,5], Ivanka Stajner[4], Raffaele Montuoro[4]

[1]Air Resources Laboratory, NOAA/OAR, College Park, MD, USA
[2]Cooperative Institute for Satellite Earth System Studies, George Mason University, Fairfax, VA, USA
[3]Earth Resources Technology, Inc.
[4]Environmental Modeling Center, NOAA/NWS/NCEP, College Park, MD, USA
[5]Lynker, Leesburg, VA, USA

Correspondence to: Wei Li (wli31@gmu.edu)

**Abstract.** Air quality forecasting system is an essential tool widely used by environmental managers to mitigate adverse health effects of air pollutants. This work presents the latest development of the next generation regional air quality model (AQM) forecast system within the Unified Forecast System (UFS) framework in the National Oceanic and Atmospheric Administration (NOAA). The UFS air quality model incorporates the U.S. Environmental Protection Agency (EPA)'s Community Multiscale Air Quality (CMAQ) model as its main chemistry component. In this system, CMAQ is integrated as a column model to solve gas and aerosol chemistry while the transport of chemical species is processed by UFS. The current AQM version 7 (AQMv7) is coupled with an earlier version of CMAQ (version 5.2.1). Here we describe the development of the updated AQMv7 by coupling to a 'state-of-the-science' CMAQ version 5.4. The updates include improvements in gas and aerosol chemistry, dry deposition processes, and structural changes to the Input/Output (IO) interface, enhancing both computational efficiency and the representation of air-surface exchange processes. A simulation was conducted for the period of June-August 2023 to assess the effects of these updates on the forecast performance of ozone ($O_3$) and fine particulate matter ($PM_{2.5}$), two major air pollutants over the continental United States (CONUS). The results show that the updated model demonstrates an enhanced capability in simulating $O_3$ over the CONUS by reducing the positive bias, leading to a reduction of the mean bias by 3%-5% and 8%-12% for hourly and the maximum daily 8-hour average $O_3$, respectively. Spatially, the updated model lowers the positive bias of hourly $O_3$ in most of the ten EPA regions, particularly within the central and northwest areas, while amplifying the $O_3$ underestimation over the sites with negative bias. Similarly, the updates induce uniformly lower fine particulate matter ($PM_{2.5}$)

concentrations across the CONUS domain, reducing the positive bias at some sites over the northeast of August and central Great Plain. The updated model does not improve model performance for PM$_{2.5}$ in the vicinity and downwind of fire emission sources, where AQMv7 shows the highest negative bias, thus indicating a focal point of model uncertainty and needed improvement. Despite these challenges, the study highlights the importance of the ongoing refinements for reliable air quality predictions from the UFS-AQM model, which is a planned future update to NOAA's current operational air quality forecast system.

## 1 Introduction

Air quality, affected by the amount and type of gaseous and particulate pollutants in the ambient air, has a wide range of impacts on human health, the ecosystem, and the economy. Criteria pollutants, such as ground-level ozone (O$_3$) and particulate matter with an aerodynamic diameter of less than 2.5 µm (PM$_{2.5}$), can cause cardiovascular and respiratory diseases (Cohen et al., 2005; Lee et al., 2014), worsen symptoms and complications of people with pre-existing health conditions (Balbus and Malina, 2009; Hooper and Kaufman, 2018), and lead to nearly 4.2 million premature deaths worldwide in 2019 with 89% of these deaths occurring in low- and middle-income countries (WHO, 2023). Acidic air pollutants, such as sulfur dioxide (SO$_2$) and nitrogen oxides (NO$_x$), can deposit onto soil and watershed and harm plant growth and aquatic life, leading to changes in ecosystems and the loss of biodiversity (Taylor et al., 1994; Lovett et al., 2009). O$_3$ can also damage forest and crop leaves and interfere with photosynthesis, resulting in yield reduction and food quality deteriorating with an estimated economic loss between 14 to 26 billion dollars globally (Van Dingenen et al., 2009; Tai et al., 2014).

To address the global concern of air pollution and alleviate its health and environmental damage, both international and national agencies play essential roles in air quality regulation and monitoring. Internationally, the World Health Organization (WHO) sets global standards for air quality and provides guidance on its health implications (WHO, 2021). The United Nations Environment Programme (UNEP) coordinates global efforts, with a specific focus on reducing short-lived climate pollutants (UNEP, 2021). In Europe, the European Environment Agency (EEA) provides information and supports the European Union's air quality efforts (EEA, 2022). In the United States, the Environmental Protection Agency (EPA) enforces the Clean Air Act and establishes national ambient air quality standards (NAAQS). Additionally, most countries maintain their own national environmental agencies, which set air quality standards and regulations tailored to local conditions. These agencies follow a comprehensive process, which includes establishing air quality standards, regulating emissions from various sources, monitoring air quality through networks of monitoring stations, and making data accessible to the public. Stringent enforcement measures are in place to ensure compliance, and research initiatives and public awareness campaigns further contribute to informed decision-making and citizen engagement. Importantly, air quality forecasts issued from some of these

agencies are an effective way to combat air pollution because accurate air pollutant predictions can protect public health by
offering advance warnings to at-risk individuals and aid in mitigation strategies by guiding industrial activities and urban planning.

The National Oceanic and Atmospheric Administration (NOAA) has taken on the responsibility of providing operational air quality forecast guidance since 2004 through the National Air Quality Forecasting Capability (NAQFC) system. The initial development of the NAQFC was based on an offline coupling between NOAA's ETA meteorological model and EPA's
Community Multiscale Air Quality (CMAQ) model, which provided $O_3$ forecast guidance over the northeast United States (Otte et al., 2005; Eder et al., 2006). Continued development and evaluation of the NAQFC enabled the system to issue $O_3$, $PM_{2.5}$, wildfire smoke and dust forecast guidance for the entire contiguous United States (CONUS), Alaska, and Hawaii in order to protect human health, the environment and economy (Mathur et al., 2008; McKeen et al., 2009; Eder et al., 2009; Stajner et al., 2012; Huang et al., 2017; Lee et al., 2017). With the National Weather Service (NWS) transition to use a new
Finite-Volume Cubed-Sphere (FV3) dynamical core in the Global Forecast System (GFS) model, in combination with GFS's improvement in data assimilation and physical parameterizations, both short and long weather forecasts are considerably improved (Harris and Lin, 2013; Zhou et al., 2019; Chen et al., 2019), which motivated s NOAA to use FV3GFS as the meteorological driver in the NAQFC (Huang et al., 2017, 2019; Chen et al., 2021). A recent version of the NAQFC, offline-coupled between version 16 of the FV3GFS (FV3GFSv16 hereafter) and CMAQv5.3.1, showed significantly different
meteorological and chemical predictions and overall improved surface $O_3$ and $PM_{2.5}$ simulations in a 72h forecast relative to its previous version (Campbell et al., 2022), and further yields similar results in a historical simulation compared with the commonly-used Weather Research & Forecasting Model (WRF; Tang et al., 2022).

In recent years, NOAA has made extensive efforts to develop the next generation weather forecast model, known as the Unified Forecast System (UFS), which is a community-based, coupled, comprehensive Earth Modeling System with the capability of
integrating a number of common components (e.g., land, ocean, atmosphere and sea ice) into different applications. The UFS framework allows for predictions that span local to global domains and range from sub-hourly to seasonal time scales (Krishnamurthy et al., 2021; Bai et al., 2023; Zhu et al., 2023). It is designed to be the unified system for NOAA's operational numerical weather prediction applications while enabling more effective collaboration among government, academia, industry, and beyond (https://ufscommunity.org, last access:30 October 2023).

The Air Quality Model (AQM; https://github.com/NOAA-EMC/AQM) is one of UFS's applications that dynamically couples the CMAQ model with the UFS weather model (https://github.com/ufs-community/ufs-weather-model) to simulate spatiotemporal variations of atmospheric composition and air quality. The chemical component in the AQM version 7 (AQMv7) is currently based on the CMAQ model version 5.2.1 (CMAQv5.2.1), which was released in 2018. Hence this version of CMAQ has become scientifically outdated, as EPA is continuously advancing the model with both scientific and

structural changes as described in Appel et al. (2021) and Murphy et al. (2021), which can potentially lead to higher biases and errors in the air quality forecast. Therefore, there is a need to update the AQMv7 to the latest version 5.4 (at the time of writing) of the CMAQ model (CMAQv5.4; https://github.com/USEPA/CMAQ/tree/5.4, last access: 30 October 2023).

The main objective of this study is to upgrade the chemical component of the current AQMv7 to the latest CMAQ model (see description in Section 2). The simulation design and model inputs are presented in Section 3. In Section 4, we compare the meteorological and air quality predictive performance between the current and updated AQMv7 (AQMv7_new hereafter) against surface observations in the CONUS. We conclude, in Section 5, that the advancement using a closer state-of-the-science chemical transport model will improve the prediction of atmospheric chemical compositions and therefore result in more accurate air quality forecasts and better protect public health across the US.

## 2 Methods: Updates to the AQM

The AQM component is a dynamic wrapper that links the UFS weather model with CMAQ through the National Unified Operational Prediction Capability (NUOPC) layer based on the Earth System Modeling Framework (ESMF). AQM has its own input and output (AQMIO) layer that can read in the online-coupled meteorology, initial and boundary conditions (IC/BC), and emissions from different sources, and then pass the updated prognostic and diagnostic chemical tracer fields back to the UFS weather model with no chemistry/aerosol feedback. CMAQ is treated as a column model for emission mapping, photolysis, gas and aerosol chemistry, dry deposition and in-cloud wet scavenging at each integration time step, while other transport terms, such as convection, advection and diffusion, are more appropriately handled in the FV3 dynamics and CCPP physics. More details of the AQMv7 structure can be found in Huang et al. (2024).

The updates of AQMv7 are mainly based on the changes from CMAQv5.2.1 to CMAQv5.4, between which there were updates for CMAQ version 5.3 (CMAQv5.3; Appel et al., 2021). The advancements of CMAQv5.3 and CMAQv5.4 are listed in its release notes for each respective version ( https://github.com/USEPA/CMAQ, last access: 30 October 2023). Here we only include the features that are used in AQMv7. The newer version usually contains various science, functionality, and computation efficiency upgrades. The following subsections describe the specifics of these changes.

### 2.1 Chemistry

Of all the three families of gas chemical mechanisms included in CMAQ, the Carbon-Bond version 6 (CB6) scheme is the most widely used for regional air quality simulations, and thus adopted in AQM. The other two chemical mechanisms currently implemented in CMAQ include Statewide Air Pollution Research Center (SAPRC) and the Regional Atmospheric Chemical Mechanism (RACM). The CB6 mechanism has evolved from revision 3 (CB6r3) in CMAQv5.2.1 to revision 5 (CB6r5) in

CMAQv5.4 (Yarwood et al., 2010; Emery et al., 2015; Yarwood et al., 2020). The associated aerosol chemistry has also been significantly updated from version 6 (AERO6) to version 7 (AERO7).

### 2.1.1 Gas chemistry

The chlorine chemistry in CB6r3 (Sarwar et al., 2012; Luecken et al., 2019) was updated in the 2019 release of CMAQv5.3, which added 5 chemical reactions and one new chlorine species compared with the previous CB6r3 mechanism in CMAQv5.2.1 (github.com/USEPA/CMAQ/blob/5.3/DOCS/Release_Notes/chlorine_chemistry_CB6r3.md, last access 21 February 2024). The same chlorine chemistry was kept in the CB6r5 mechanism. Both detailed and simplified bromine and iodine chemistry schemes (Sarwar et al., 2015) are implemented in CMAQ, the latter of which is used in AQMv7 to reduce the computational demand. The simple halogen chemistry uses a first-order constant to calculate the $O_3$ loss rate to seawater as a function of atmospheric pressure. With the updates of the detailed halogen chemistry (Sarwar et al., 2019), the $O_3$ loss rate constant has been recalculated in CMAQv5.3 and further rederived in CMAQv5.4 with an increased and decreased value relative to its previous version, respectively. The final result is a reduction of $O_3$ in the ocean (https://github.com/USEPA/CMAQ/blob/5.3/DOCS/Release_Notes/simple_halogen_chemistry.md, last access: 22 March 2024). Other chemistry changes in CB6r5 (Burkholder et al., 2019) include updates in reaction rate constants, reaction products and yields, photolysis rates of some species, and the addition of new reactions. The overall impacts of the mechanism migration from CB6r3 to CB6r5 are marginal increases in both summer and winter months (https://github.com/USEPA/CMAQ/wiki/CMAQ-Release-Notes:-Chemistry:-Carbon-Bond-6-Mechanism-(CB6), last access: 31 October 2023).

### 2.1.2 Aerosol chemistry

AERO7 has extensive changes from AERO6 incorporating a number of key improvements, such as updating the yields of monoterpene secondary organic aerosol (SOA) resulting from the photooxidation by hydroxyl radicals (OH) and $O_3$ (Saha and Grieshop, 2016), adding the formation and subsequent partitioning of organic nitrate (Pye et al., 2015), introducing the inclusion of water uptake on hydrophilic organic compounds as described in Pye et al. (2017), accounting for the consumption of inorganic sulfate during the formation of isoprene epoxydiol (IEPOX) organosulfates (Pye et al., 2013; Zhang et al., 2018b), and enhancing computational efficiency by replacing the Odum two-product fit (Odum et al., 1996; Henze and Seinfeld, 2006; Carlton et al., 2010) with a new parameterization of anthropogenic SOA yields through a volatility basis set (VBS) approach (Pye et al., 2010, 2019). The updated monoterpene oxidation yield in the VBS fit and the inclusion of water uptake in AERO7 will generally increase organic aerosol and PM2.5 primarily in the vegetated southeast US during summertime (Xu et al., 2018; Zhang et al., 2018a), the latter of which will also affect deposition and aerosol optical depth (AOD) by modulating aerosol size (Pye et al., 2017).

## 2.2 Dry deposition

There are two air-surface exchange models starting from CMAQv5.3: the Models-3 dry (M3Dry) deposition model and the
Surface Tiled Aerosol and Gaseous Exchange (STAGE) model. Currently, only M3Dry is adopted in AQMv7. Some important updates have been made for $O_3$ and aerosol deposition depending on land use types since the release of CMAQv5.2.1. The $O_3$ dry deposition resistance to snow was raised by 10 times from 1000 to 10 000 s m$^{-1}$ following the observed evidence in Helmig et al (2007), leading to a significant increase of ambient $O_3$ over snow-covered regions. The ground $O_3$ resistance over soil has also been modified to be dependent on soil moisture (Mészáros et al., 2009; Fares et al., 2012) with a generally decreased
value relative to the previous dry deposition scheme and thus result in more $O_3$ depositing to the soil surface and less remaining in the ambient air.

The aerosol dry deposition scheme has been updated in both CMAQv5.3 and CMAQv5.4. The revised parameterization of aerosol dry deposition in CMAQv5.3 added a leaf area index (LAI) factor in the boundary layer resistance to account for large depositions over forest canopies, which greatly reduces the coarse-mode particle dry deposition velocity (Shu et al., 2022;
Appel et al., 2021). The scheme is further improved in CMAQv5.4 by introducing a two-term impaction efficiency to represent macroscale and microscale obstacles, which differ by land use categories including needleleaf forest, broadleaf forest, and grassland (Pleim et al., 2022). The most significant changes of mass dry deposition velocity are found for the accumulation mode over the forested areas with an increase by almost an order of magnitude, causing an overall reduced $PM_{2.5}$ in the continuous US relative to CMAQv5.3.

## 2.3 Structural changes

A number of changes have been made to the Input/Output (IO) framework of CMAQ (Figure 1). Emission reading, mapping, and scaling are controlled in the Detailed Emissions Scaling, Isolation, and Diagnostic (DESID) module in CMAQv5.3 and beyond. The module can read any number of offline gridded and point emission files by their sources (defined as streams) and apply scaling factors on a per-species and per-region basis for each stream, allowing users to perform emission scaling and
perturbation tests with great ease and flexibility (Murphy et al., 2021). The opening, description, extraction, and interpolation of the meteorological and emission variables are encapsulated in the centralized I/O (CIO) module from CMAQv5.3, lowering computational memory requirements and easing code maintenance. The Introduction of the Explicit and Lumped air quality Model Output (ELMO) module is included in CMAQv5.4, which can synthesize the definition, calculation, and maintenance of individual or aggregate gas and particulate matter parameters (e.g., $PM_{2.5}$) online, saving time and storage to run post-
processing tools. Implementing these changes requires new control name lists and extensive code updates in AQMv7_new.

**Figure 1: Summary of the IO changes in the AQMv7_new model. Three major structural changes are highlighted in red.**

## 3 Simulation design and evaluation protocol

Despite the chemistry and dry deposition updates described in the last section, other model components and configurations are

the same in order to isolate the model performance changes caused by the updates. Table 1 summarizes the model domain, physical settings and emission inputs, as well as some additional information.

**Table 1: UFS-AQM model components and configurations. The abbreviation N/A stands for not applicable in this table.**

| Model attributes | Configuration | Reference |
|---|---|---|
| Domain | North America Cantered on 50° N 118° W | N/A |
| Horizontal resolution | 13km | N/A |
| Vertical resolution | 64 levels from near the surface up to the top of the stratosphere | N/A |
| Meteorological ICs and BCs | FV3GFSv16.3 | https://nws.weather.gov/ (last access: 25 November 2023) |

| | | |
|---|---|---|
| Chemical ICs and BCs | Static monthly AM4 for gases and aerosol species and GEFS-Aerosol for dynamic smoke and dust | Horowitz et al. (2020); Tang et al. (2021); Lin et al,. (2020, 2024) |
| Microphysics | GFDL six-category cloud microphysics scheme | Lin et al. (1983); Lord et al. (1984); Krueger et al. (1995); Chen and Lin (2011, 2013) |
| PBL physics scheme | sa-TKE-EDMF | Han and Bretherton (2019) |
| Shallow and deep cumulus parameterization | SAS scheme | Han and Pan (2011); Han et al. (2017) |
| Shortwave and longwave radiation | RRTMg | Mlawer et al. (1997); Clough et al. (2005); Iacono et al. (2008) |
| Land surface model | Noah land surface model with 20-category IGBP land cover | Chen and Dudhia (2001); Ek et al. (2003); Tewari et al. (2004) |
| Surface layer | Monin-Obukhov | Monin and Obukhov (1954); Grell et al. (1994); Jimenez et al. (2012) |
| Anthropogenic emissions (CONUS) | Area Sources: NEIC2016v1 Point Sources: NEIC2016v1 with Briggs plume rise | NEI (2019); Briggs (1965) |
| Anthropogenic emissions (Outside CONUS) | CEDSv2; HTAPv2.2; OMI-HTAP $SO_2$ 2019 | O'Rourke et al. (2021); Janssens-Maenhout et al. (2015); Liu et al. (2018) |
| Biogenic emissions | MEGAN2.1 driven by GFSv16 meteorology | Guenther et al. (2012) |
| Wildfire emissions | RAVE with Sofiev plume rise | Li et al., (2022); Sofiev et al. (2012) |
| Other inline/Offline emissions | FENGSHA windblown dust scheme | Fu et al. (2014); Huang et al. (2015); Dong et al. (2016) |
| | Sea spray emissions | Kelly et al. (2010); Gantt et al. (2015) |

The model domain covers North America (NA) with a horizontal resolution of ~13km and 64 vertical layers spanning from
the surface up to the top of the stratosphere (~ 0.4 hPa). The Common Community Physics Package (CCPP) FV3GFSv16.3
physics suite (Heinzeller et al., 2023) is used to provide meteorological conditions, where its physical configurations include
the Monin-Obukhov Similarity surface layer (Monin and Obukhov, 1954; Grell et al., 1994; Jiménez et al., 2012), the Noah
land surface scheme (Chen and Dudhia, 2001; Ek et al., 2003; Tewari et al., 2004), the Rapid Radiative Transfer Model
(RRTM) longwave and shortwave radiation schemes (Mlawer et al., 1997; Clough et al., 2005; Iacono et al., 2008), the
Simplified Arakawa Schubert (SAS) cumulus parameterization (Han and Pan, 2011; Han et al., 2017), the Geophysical Fluid
Dynamics Laboratory (GFDL) six-category cloud microphysics scheme (Lin et al., 1983; Lord et al., 1984; Krueger et al.,
1995; Chen and Lin, 2011, 2013), and the sa-TKE-EDMF planetary boundary layer (PBL) scheme (Han and Bretherton, 2019).

Anthropogenic emissions outside of the CONUS are from CEDSv2-2019 for all gases, except for sulfur dioxide ($SO_2$) only in the ocean, organic carbon (OC), and black carbon (BC) (Table 1). The blended Ozone Monitoring Instrument-HTAP (OMI-HTAP) 2019 dataset (https://so2.gsfc.nasa.gov/measures.html, last access: 15 March 2024) provides $SO_2$ emissions over land, and the emissions of coarse particulate matter (PMC) and $PM_{2.5}$ are from HTAPv2-2010. Within the CONUS, all gas and aerosol anthropogenic emissions are from the National Emissions Inventory Collaborative (NEIC) 2016 version 1 (2016v1). The NEIC2016v1 provides both area and point emissions, the latter of which is further calculated inline in AQM using the Briggs plume rise method. The same plume rise method is also applied to the wildfire emissions from the Regional ABI and VIIRS fire Emissions version 1 (RAVE1) inventory, in which all gaseous emissions are scaled from CO and speciated particulate matter emissions are scaled from total $PM_{2.5}$. Both the windblown dust and sea salt emissions are calculated inline. The dust scheme is based on a novel FENGSHA model (Fu et al., 2014; Huang et al., 2015; Dong et al., 2016), which is dependent on the land cover, soil type, soil moisture, and friction velocity. Biogenic emissions are from the Model of Emissions of Gases and Aerosols from Nature version 2.1 (MEGAN2.1) driven by the GFSv16 meteorology. The area source anthropogenic and biogenic emissions are both processed and calculated inline using the NOAA Emissions and eXchange Unified System (NEXUS) component (Campbell et al., 2020), which is based upon the Harmonized Emissions Component (HEMCO) 3.0 (Lin et al., 2021). The chemical initial and boundary conditions (ICs/BCs) are from the monthly mean Atmospheric Model version 4 (AM4) outputs for gas and aerosol species with additional dynamic BCs for dust and smoke aerosols from the aerosol forecast member in the Global Ensemble Forecast System (GEFS-Aerosols), which can better capture the aerosol intrusion events from outside of the domain and thus improve the prediction of air quality (Tang et al., 2021).

The simulations for both AQMv7 and AQMv7_new were performed for three months from June to August of 2023, during which there were extensive wildfire activities over the northwest U.S. and Canada. The air quality observations from the EPA AirNow network are used to evaluate the model performance and the evaluation is conducted using the publicly available software MELODIES-MONET (Model EvaLuation using Observations, DIagnostics and Experiments Software (MELODIES) with the Model and ObservatioN Evaluation Toolkit; Baker and Pan, 2017; https://melodies-monet.readthedocs.io/en/stable/, last access: 29 October 2024). The software can produce flexible diagnostic assessments by pairing models and observations, plotting spatial maps, and calculating statistics such as mean bias (MB), normalized mean bias (NMB), median bias (MdnB), normalized median bias (NMdnB), mean absolute error (MAE), normalized mean error (NME), coefficient of determination ($R^2$), root-mean-square error (RMSE), and the index of agreement (IOA). A meteorological evaluation was also conducted using the U.S. EPA Atmospheric Model Evaluation Tool (AMET; Appel et al., 2011; https://www.cmascenter.org/amet/, last access: 15 March 2024) against the observations collected from the Surface Weather Observations and Reports for Aviation Routine Weather Reports (METAR) and Earth System Research Laboratory's (ESRL's) Radiosonde Database (RAOB).

## 4 Results: Assessment and evaluation of updates

In this section, we compared the performance of the current and updated models in their capability of predicting summer season (June, July, and August 2023) $O_3$ and $PM_{2.5}$ as they are the most important air pollutants of concern. Although both models are driven by the same CCPP GFSv16 physics suite, we first briefly evaluated the simulation of some meteorological factors critical for $O_3$ and $PM_{2.5}$ formation and transport, which can provide insights into the overall model performance in air quality predictions.

## 4.1 Meteorology evaluation

Figure 2 shows the anomaly correlation coefficient (ACC) and mean bias (MB) between four simulated and observed variables at each site in August, including 2m temperature (TEMP2) and specific humidity (Q2), 10m wind speed (WS10) and direction (WD10) with more statistics listed in Table 2. Similar spatial patterns of ACC and MB are found for June and July (Figure S1-2). Some diurnal variation and vertical distribution comparisons were also conducted and shown in Figure S3-S11. TEMP2, Q2 and WS10 in the CONUS are well simulated with high correlation coefficients (CORR) of 0.93-0.95, 0.91-0.93 and 0.56-0.65 and low mean bias of -0.03 - -0.56 °C, -0.81 - -1.41 g kg$^{-1}$, and -0.15 - -0.25 m s$^{-1}$, respectively (Table 2). While cold biases are found in the northeastern and western US (Figure 2 and S1-2) at the surface mainly driven by nighttime underpredictions (Figure S3-8), the vertical distribution shows a nationwide warm bias (Figure S9-11). Specific humidity has a universal dry bias within the domain both at the surface and vertically with the latter showing higher bias up to 10 g kg$^{-1}$ at some sites. Such biases in TEMP2 and Q2 suggest an overly stable atmosphere in the GFSv16 physics during summer, which may influence overpredictions in trace gases in the lowest model layers by supressing advection and diffusion. The diurnal evaluations also indicate overpredictions in TEMP2 during the daytime both in the western and eastern US, where the warm and dry biases may further exacerbate $O_3$ formation and overpredictions, especially in the eastern U.S. (See Section 4.2 below). Furthermore, WS10 is underestimated in the western and part of eastern US by up to 3 m/s, which also contributes to the overestimation of $O_3$ therein due to reduced dilution. The WD10 demonstrated relatively worse predictions, especially in its vertical distributions, with low CORR values smaller than 0.6 and a high mean bias greater than 20° at most sites, adding more uncertainties to the transport of pollutants in addition to those from WS10. AMET accounts for the wind direction vector issue in its calculation of the evaluation statistics.

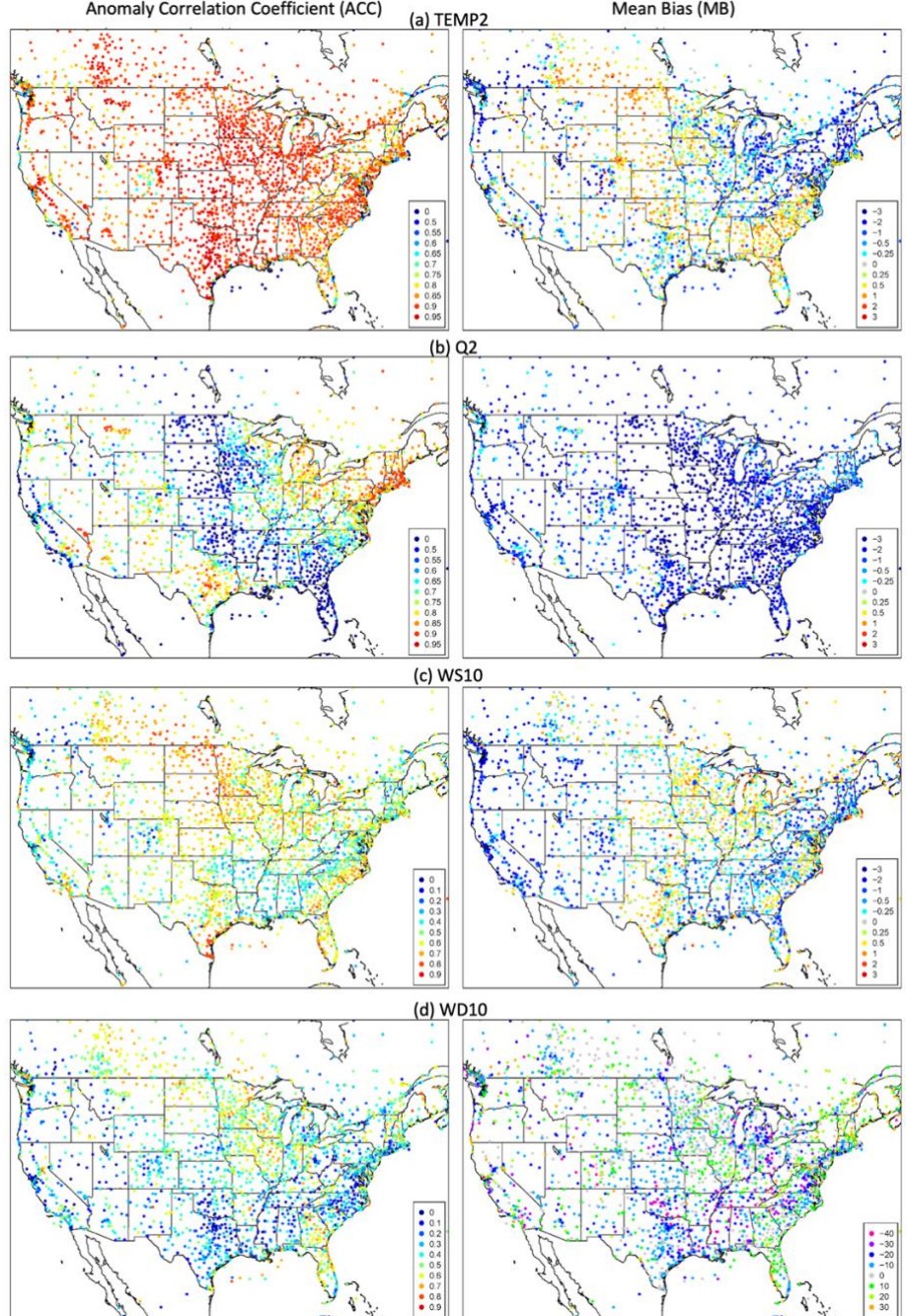

**Figure 2: Anomaly correlation coefficient (ACC; left column) and mean bias (MB) between GFSv16.3 simulated and observed TEMP2 (a; °C), Q2(b; g/kg), WS10 (c; m/s), and WD10 (d; degree) at the surface in August 2023.**

**Table 2: Evaluation statistics of TEMP2 (K), Q2(g/kg) and WS10 (m/s) from the GFSv16.3 simulation in summer months of 2023. MB, MAE and RMSE have the same units as the variables.**

| Region | Variable | Month | CORR | ACC | MB | NMB (%) | MAE | NME (%) | RMSE |
|---|---|---|---|---|---|---|---|---|---|
| CONUS | TEMP2 | June | 0.93 | 0.93 | -0.56 | -1.05 | 1.93 | 3.66 | 2.52 |
| | | July | 0.93 | 0.93 | -0.44 | -0.83 | 2.00 | 3.76 | 2.64 |
| | | August | 0.95 | 0.95 | -0.03 | -0.06 | 1.73 | 3.27 | 2.29 |
| | Q2 | June | 0.91 | 0.89 | -0.81 | -2.65 | 1.43 | 4.70 | 1.94 |
| | | July | 0.93 | 0.90 | -1.21 | -4.29 | 1.63 | 5.76 | 2.12 |
| | | August | 0.92 | 0.87 | -1.41 | -4.72 | 1.70 | 5.66 | 2.25 |
| | WS10 | June | 0.59 | 0.51 | -0.17 | -0.75 | 1.30 | 5.83 | 1.76 |
| | | July | 0.56 | 0.47 | -0.25 | -1.00 | 1.36 | 5.35 | 1.85 |
| | | August | 0.65 | 0.56 | -0.15 | -0.57 | 1.24 | 4.72 | 1.68 |
| Eastern US (100°W east) | TEMP2 | June | 0.93 | 0.93 | -0.60 | -1.15 | 1.88 | 3.69 | 2.44 |
| | | July | 0.92 | 0.92 | -0.32 | -0.64 | 1.86 | 3.59 | 2.43 |
| | | August | 0.95 | 0.95 | 0.00 | 0.00 | 1.60 | 3.19 | 2.08 |
| | Q2 | June | 0.89 | 0.86 | -0.88 | -2.91 | 1.56 | 5.15 | 2.07 |
| | | July | 0.86 | 0.78 | -1.55 | -5.47 | 1.95 | 6.87 | 2.48 |
| | | August | 0.89 | 0.81 | -1.65 | -5.55 | 1.91 | 6.40 | 2.45 |
| | WS10 | June | 0.59 | 0.50 | 0.02 | 0.08 | 1.19 | 5.34 | 1.59 |
| | | July | 0.53 | 0.44 | 0.00 | -0.01 | 1.22 | 6.09 | 1.63 |
| | | August | 0.63 | 0.54 | 0.05 | 0.19 | 1.12 | 4.78 | 1.50 |
| Western US (100°W west) | TEMP2 | June | 0.93 | 0.93 | -0.45 | -0.86 | 2.05 | 4.03 | 2.70 |
| | | July | 0.93 | 0.93 | -0.08 | -0.17 | 2.32 | 4.44 | 3.04 |
| | | August | 0.93 | 0.93 | 0.02 | 0.06 | 2.07 | 4.12 | 2.76 |
| | Q2 | June | 0.88 | 0.86 | -0.60 | -2.36 | 1.07 | 4.20 | 1.54 |
| | | July | 0.80 | 0.75 | -1.04 | -3.96 | 1.49 | 5.66 | 2.03 |
| | | August | 0.83 | 0.77 | -0.98 | -3.48 | 1.37 | 4.87 | 1.83 |
| | WS10 | June | 0.61 | 0.53 | -0.55 | -2.54 | 1.52 | 7.05 | 2.06 |
| | | July | 0.56 | 0.48 | -0.44 | -1.71 | 1.53 | 6.02 | 2.05 |
| | | August | 0.63 | 0.54 | -0.48 | -1.86 | 1.41 | 5.47 | 1.89 |

255

## 4.2 O$_3$ evaluation

Figure 3 displays the spatial maps of hourly O$_3$ distribution in the CONUS averaged in June-August 2023 from two model simulations and AirNow observations, as well as the model mean bias at each site. The western US generally has a higher level of O$_3$ relative to the eastern US, reflecting the overall O$_3$ spatial distribution during summertime. The AQMv7 captures this spatial pattern, yet with a positive bias at the majority of the AirNow sites. A higher positive bias of more than 20 ppb can be found near the west and east coast compared to the smaller or negative bias in other regions for all the months, indicating the land-sea interactions may not be well represented in the model. The relatively large O$_3$ overestimates are also impacted by the near-surface meteorological biases described previously (i.e., too warm and dry during the day and too cool and dry at night), as well as an overly stable boundary layer. The most noticeable negative bias can be seen in the northeastern US of June, which is attributable to the record-breaking wildfire smoke transported from Quebec. This indicates the contributions from fires to O$_3$ enhancements are underestimated in the model. The AQMv7_new model shows a nation-wise decrease in O$_3$ mixing ratios, which reduces the high positive bias over the coastal sites. This reduction, however, also exacerbates the O$_3$ underestimation for the sites with negative bias.

Averaging across the CONUS, the hourly O$_3$ time series from the AQMv7 simulation (blue line in Figure 4) show that the model captures the temporal variation with an R$^2$ value of 0.44, 0.50 and 0.49 from June to August, respectively (Table 3-5). However, except for the fire-related O$_3$ underestimation in June, the model overestimates both the peak values at noon and the low values at night with a mean bias of 1.54 ppb (4.23%) in June, 4.84 ppb (14.55%) in July, and 7.21 ppb (23.12%) in August, which explains the high positive bias shown in Figure 2. Such overestimation of O$_3$ is mitigated, especially during nighttime, by the updated model, reducing the mean bias by 3-5%. The RMSE and IOA values of hourly O$_3$ are also improved by the model updates, indicating an enhanced model performance in simulating O$_3$ in the CONUS overall. We also evaluated the model performance of the maximum daily 8-hour average (MDA8) O$_3$ simulation in Figure 4 with the statistics listed in Table S1-3. The AQMv7 model underestimates MDA8 O$_3$ by 1.87 ppb (3.82%) in June while overpredicts it by 1.94 ppb (4.21%) and 5.18 ppb (11.89%) in July to August, respectively. The reduction effects in AQMv7_new lowers the positive bias in July and August by 0.15 ppb (8%) and 0.62 (12%) and amplifies the negative bias in June by 0.23 ppb (12%). Considering the underestimation of daytime O$_3$ in AQMv7 due to the big impact from fire in June, this may indicate that the model updates can improve summertime MDA8 O$_3$ simulation when influences from fire are small.

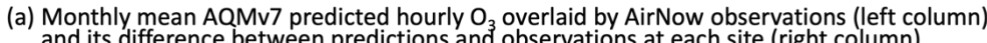

(a) Monthly mean AQMv7 predicted hourly O₃ overlaid by AirNow observations (left column) and its difference between predictions and observations at each site (right column)

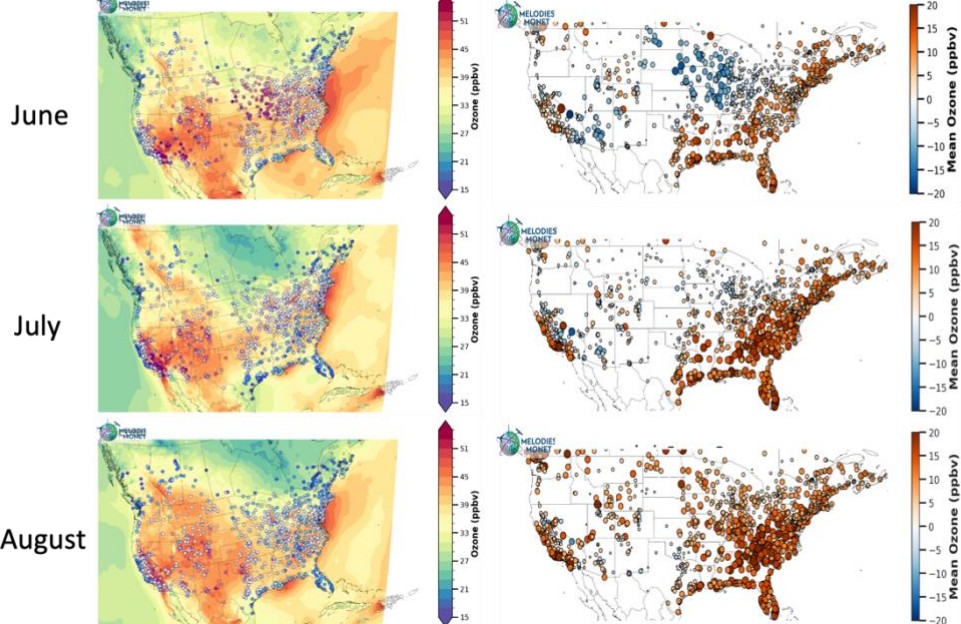

(b) Monthly mean AQMv7_new predicted hourly O₃ overlaid by AirNow observations (left column) and its difference between predictions and observations at each site (right column)

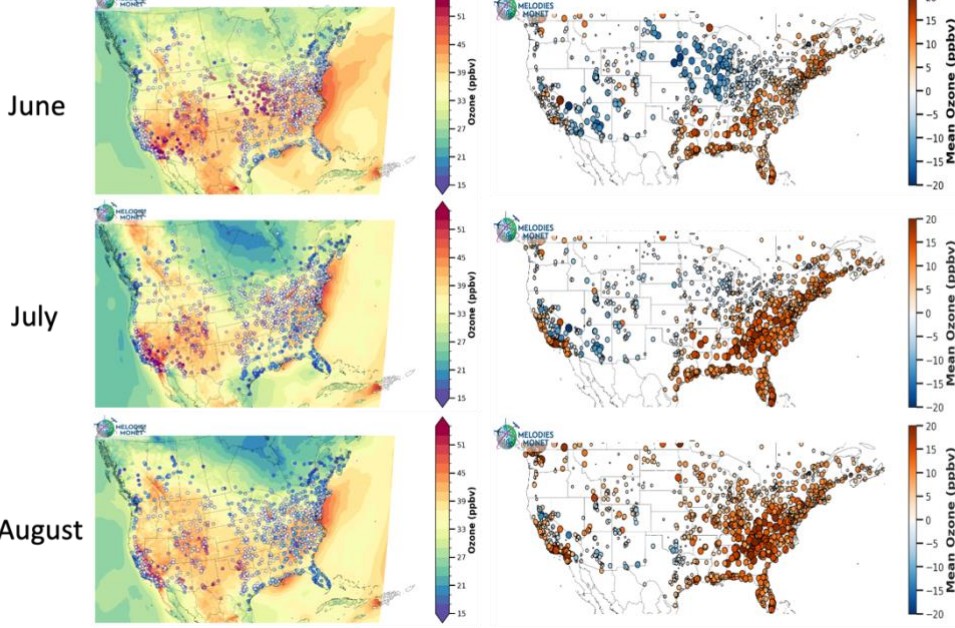

**Figure 3: Maps of monthly mean hourly O₃ in the CONUS predicted by AQMv7 (a) and AQMv7_new (b) overlaid by AirNow observation sites (left column) and its bias between simulations and observations (model - AirNow) at each site (right column).**

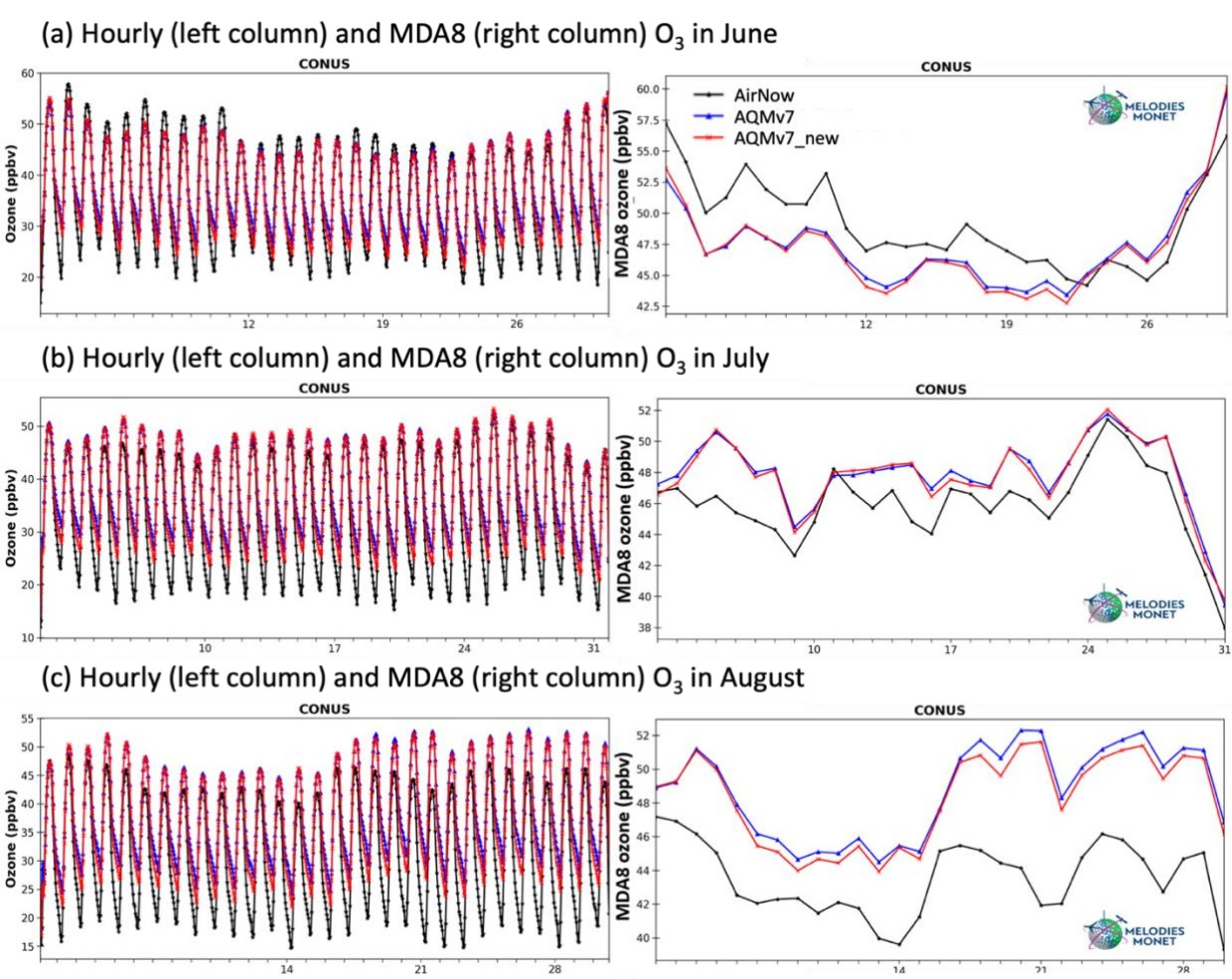

Figure 4: Time series of hourly (left column) and MDA8 (right column) O₃ in the CONUS from AirNow observations (black line), AQMv7 (blue line) and AQMv7_new (red line) predictions for June (a), July (b) and August (c) 2023.

Table 3: Hourly O₃ evaluation statistics of the AQMv7 and AQMv7_new simulations against the AirNow network in the CONUS and different regions in June 2023. The bold numbers in AQMv7_new indicate an improvement relative to those in AQMv7.

| Region | Model | MB (ppb) | NMB (%) | MdnB (ppb) | NMdnB (%) | $R^2$ | RMSE (ppb) | IOA |
|---|---|---|---|---|---|---|---|---|
| CONUS | AQMv7 | 1.54 | 4.23 | 1.26 | 3.60 | 0.44 | 12.91 | 0.78 |
| | AQMv7_new | **0.37** | **1.01** | **-0.11** | **-0.31** | **0.46** | **12.71** | **0.80** |
| | AQMv7 | 7.04 | 23.15 | 6.25 | 21.54 | 0.46 | 13.10 | 0.76 |

| Region | Model | MB (ppb) | NMB (%) | MdnB (ppb) | NMdnB (%) | R² | RMSE (ppb) | IOA |
|---|---|---|---|---|---|---|---|---|
| Region 1 (northeast) | AQMv7_new | **6.10** | **19.77** | **4.90** | **16.91** | 0.45 | **12.87** | **0.78** |
| Region 2 (NY-NJ) | AQMv7 | 4.63 | 14.04 | 4.07 | 12.71 | 0.44 | 12.98 | 0.79 |
| | AQMv7_new | **4.22** | **12.79** | **3.47** | **10.84** | **0.45** | 12.99 | **0.80** |
| Region 3 (mid-Atlantic) | AQMv7 | 2.83 | 7.53 | 1.95 | 5.13 | 0.42 | 13.14 | 0.77 |
| | AQMv7_new | **2.32** | **6.17** | **1.25** | **3.30** | **0.44** | **12.89** | **0.78** |
| Region 4 (southeast) | AQMv7 | 6.43 | 18.63 | 6.22 | 18.30 | 0.46 | 14.17 | 0.73 |
| | AQMv7_new | **5.67** | **16.40** | **5.14** | **10.65** | 0.48 | **13.54** | **0.76** |
| Region 5 (upper Midwest) | AQMv7 | -4.29 | -10.40 | -5.04 | -12.61 | 0.45 | 14.57 | 0.75 |
| | AQMv7_new | -5.10 | -12.36 | -5.87 | -14.68 | **0.48** | **14.48** | **0.77** |
| Region 6 (south) | AQMv7 | 4.62 | 13.69 | 4.79 | 14.50 | 0.54 | 12.67 | 0.81 |
| | AQMv7_new | **3.24** | **9.60** | **3.21** | **9.48** | 0.55 | **12.07** | **0.83** |
| Region 7 (central Great Plain) | AQMv7 | -6.39 | -13.94 | -7.52 | -15.99 | 0.42 | 15.08 | 0.71 |
| | AQMv7_new | -7.38 | -16.12 | -8.56 | -18.22 | **0.45** | 15.15 | **0.73** |
| Region 8 (northern Great Plain) | AQMv7 | -2.20 | -5.23 | -2.97 | -6.90 | 0.27 | 12.15 | 0.68 |
| | AQMv7_new | -3.93 | -9.32 | -4.54 | -10.55 | **0.33** | **12.07** | **0.71** |
| Region 9 (southwest) | AQMv7 | 0.05 | 0.12 | 0.04 | 0.10 | 0.59 | 10.01 | 0.85 |
| | AQMv7_new | -2.08 | -5.41 | -2.09 | -5.65 | 0.59 | 10.24 | **0.86** |
| Region 10 (northwest) | AQMv7 | 0.82 | 2.84 | -0.02 | -0.06 | 0.49 | 9.32 | 0.82 |
| | AQMv7_new | **0.04** | **0.12** | -1.05 | -3.37 | 0.49 | 9.47 | **0.83** |


**Table 4: Same as Table 3 but for July 2023.**

| Region | Model | MB (ppb) | NMB (%) | MdnB (ppb) | NMdnB (%) | R² | RMSE (ppb) | IOA |
|---|---|---|---|---|---|---|---|---|
| CONUS | AQMv7 | 4.84 | 14.55 | 4.14 | 12.55 | 0.50 | 12.77 | 0.80 |
| | AQMv7_new | **3.56** | **10.71** | **2.86** | **8.68** | 0.50 | **12.53** | **0.82** |
| Region 1 (northeast) | AQMv7 | 8.11 | 24.67 | 7.36 | 23.01 | 0.51 | 13.84 | 0.78 |
| | AQMv7_new | **7.68** | **23.36** | **6.70** | **20.93** | 0.50 | 14.00 | 0.78 |
| Region 2 (NY-NJ) | AQMv7 | 5.88 | 16.89 | 5.06 | 14.47 | 0.49 | 13.41 | 0.80 |
| | AQMv7_new | **5.70** | **16.35** | **4.79** | **13.68** | **0.50** | 13.51 | **0.81** |
| Region 3 (mid-Atlantic) | AQMv7 | 7.49 | 21.95 | 6.25 | 17.87 | 0.45 | 14.15 | 0.74 |
| | AQMv7_new | **7.17** | **21.01** | **6.03** | **17.23** | **0.46** | **14.01** | **0.76** |

| Region | Model | MB (ppb) | NMB (%) | MdnB (ppb) | NMdnB (%) | R² | RMSE (ppb) | IOA |
|---|---|---|---|---|---|---|---|---|
| Region 4 (southeast) | AQMv7 | 11.65 | 42.50 | 11.12 | 41.17 | 0.51 | 15.88 | 0.70 |
| | AQMv7_new | **10.81** | **39.44** | **10.17** | **37.67** | **0.53** | **15.16** | **0.73** |
| Region 5 (upper Midwest) | AQMv7 | 0.91 | 2.58 | -0.18 | -0.49 | 0.47 | 11.54 | 0.80 |
| | AQMv7_new | **0.06** | **0.18** | -1.09 | -3.02 | **0.49** | **11.34** | **0.81** |
| Region 6 (south) | AQMv7 | 6.53 | 21.85 | 7.15 | 25.53 | 0.65 | 12.05 | 0.83 |
| | AQMv7_new | **4.49** | **15.03** | **5.08** | **18.16** | 0.63 | **11.22** | **0.85** |
| Region 7 (central Great Plain) | AQMv7 | 2.15 | 5.97 | 0.81 | 2.18 | 0.41 | 11.27 | 0.76 |
| | AQMv7_new | **0.39** | **1.09** | -0.95 | -2.56 | **0.43** | **10.85** | **0.79** |
| Region 8 (northern Great Plain) | AQMv7 | 0.45 | 1.12 | -0.59 | -1.43 | 0.42 | 10.28 | 0.78 |
| | AQMv7_new | -1.97 | -4.94 | -2.80 | -6.82 | **0.47** | **10.06** | **0.80** |
| Region 9 (southwest) | AQMv7 | 1.57 | 5.55 | 0.80 | 3.08 | 0.62 | 8.62 | 0.88 |
| | AQMv7_new | **0.51** | **1.78** | **-0.51** | **-1.95** | 0.60 | 8.83 | 0.88 |
| Region 10 (northwest) | AQMv7 | -2.06 | -19.25 | -0.20 | -2.65 | 0.21 | 13.95 | 0.52 |
| | AQMv7_new | -2.83 | -26.39 | -0.80 | -10.56 | 0.20 | 14.21 | 0.49 |

**Table 5: Same as Table 3 but for August 2023.**

| Region | Model | MB (ppb) | NMB (%) | MdnB (ppb) | NMdnB (%) | R² | RMSE (ppb) | IOA |
|---|---|---|---|---|---|---|---|---|
| CONUS | AQMv7 | 7.21 | 23.12 | 6.45 | 21.51 | 0.49 | 13.50 | 0.78 |
| | AQMv7_new | **5.65** | **18.11** | **4.79** | **15.98** | 0.49 | **12.87** | **0.80** |
| Region 1 (northeast) | AQMv7 | 7.31 | 26.10 | 6.47 | 22.32 | 0.44 | 12.07 | 0.74 |
| | AQMv7_new | **6.63** | **23.68** | **5.69** | **19.61** | **0.46** | **11.70** | **0.76** |
| Region 2 (NY-NJ) | AQMv7 | 6.96 | 23.50 | 6.10 | 20.35 | 0.46 | 12.70 | 0.77 |
| | AQMv7_new | **6.59** | **22.24** | **5.47** | **18.24** | **0.47** | 12.70 | **0.78** |
| Region 3 (mid-Atlantic) | AQMv7 | 9.26 | 29.51 | 8.01 | 25.04 | 0.42 | 14.48 | 0.72 |
| | AQMv7_new | **8.56** | **27.56** | **7.41** | **23.14** | **0.43** | **14.23** | **0.73** |
| Region 4 (southeast) | AQMv7 | 12.97 | 46.99 | 12.19 | 45.16 | 0.50 | 16.84 | 0.69 |
| | AQMv7_new | **11.77** | **42.64** | **10.96** | **40.58** | 0.52 | **15.88** | **0.72** |
| Region 5 (upper Midwest) | AQMv7 | 6.85 | 21.99 | 5.94 | 19.17 | 0.48 | 12.56 | 0.78 |
| | AQMv7_new | **5.72** | **18.36** | **4.59** | **14.82** | **0.50** | **11.93** | **0.80** |
| Region 6 (south) | AQMv7 | 4.65 | 13.44 | 4.78 | 14.05 | 0.66 | 11.79 | 0.85 |
| | AQMv7_new | **2.24** | **6.48** | **2.39** | **7.03** | 0.65 | **11.00** | **0.87** |

| | | | | | | | | |
|---|---|---|---|---|---|---|---|---|
| Region 7 (central Great Plain) | AQMv7 | 8.17 | 25.16 | 7.30 | 22.12 | 0.48 | 13.02 | 0.75 |
| | AQMv7_new | **6.11** | **18.81** | **5.08** | **15.41** | 0.48 | **11.92** | **0.78** |
| Region 8 (northern Great Plain) | AQMv7 | 4.34 | 11.38 | 3.38 | 8.40 | 0.35 | 12.25 | 0.72 |
| | AQMv7_new | **1.28** | **3.36** | **0.37** | **0.96** | **0.42** | **10.86** | **0.78** |
| Region 9 (southwest) | AQMv7 | 5.80 | 16.14 | 5.18 | 15.24 | 0.56 | 13.10 | 0.82 |
| | AQMv7_new | **3.35** | **9.32** | **2.54** | **7.48** | **0.57** | **12.27** | **0.85** |
| Region 10 (northwest) | AQMv7 | 6.02 | 20.79 | 4.63 | 16.54 | 0.54 | 12.44 | 0.82 |
| | AQMv7_new | **4.80** | **16.59** | **3.11** | **11.12** | 0.52 | 12.45 | **0.83** |

In addition to the statistics listed in Table 3-5, hit rate, false alarm rate, and critical success index (CSI) are metrics commonly used to evaluate the performance of predictions, providing valuable insights into different aspects of forecast accuracy and reliability. Figure 5 compares these three metrics between AQMv7 and AQMv7_new at different hourly $O_3$ thresholds across the CONUS. Although both models have difficulties in predicting higher levels of $O_3$ indicated by the decrease of hit rate and CSI and the increase of false alarm rate as the threshold changes from 0 ppbv to 100 ppbv, the new model yields a higher CSI

value when $O_3$ is greater than 60 ppb for the three months. However, it also exhibits a slightly lower hit rate and higher false alarm rate at the 60 ppbv threshold, especially for June and July. This suggests that while the new model is more successful in accurately predicting significant ozone events, it does so less frequently with a higher number of false positives in the upper concentration ranges (e.g., > 80 ppb). However, the new model can better capture the moderate $O_3$ concentration ranges near 40 ppb as indicated by the higher hit rate and lower false alarm rate with a similar accuracy (CSI value) for all the months.

We also assessed the model simulations in each of the 10 EPA regions (R1-R10 hereafter) in Figure 6a and Table 3-5 to further examine how the updates will affect the model performance regionally. Except for the underestimation in the upper Midwest (R5) and Great Plain (R7-8) in June and the northwest (R10) in July, the AQMv7 model overestimates hourly $O_3$ in all regions with the highest mean bias value found in the northeast (R1) for June (7.04 ppb) and the southeast (R4) for July and August (11.65 ppb and 12.97 ppb, respectively). Compared to the AQMv7 model, the statistical distributions of hourly $O_3$ from the

AQMv7_new model move to the lower end, which reduces the positive bias in most of the regions by 0.18 ppb – 3.06 ppb (3.06% - 95.12%) as indicated by the improved statistics (bold numbers) in Table 3-5. Interestingly, the central and southwest regions (R6-R9) have a higher sensitivity to the model updates relative to other regions in all three months, which is likely due to the combined effects of $O_3$ chemistry and dry deposition. As described in Section 2, the halogen chemistry updates reduce $O_3$ over sea water, which can be transported into the central U.S. dominated by southerly winds in summer, such as the Great

Plain low-level jet (Zhu and Liang, 2013; Li et al., 2020). In addition, the added dependence of $O_3$ dry deposition velocity to soil moisture leads to more $O_3$ uptake by dry soil than wet soil (Appel et al., 2021) and the central and western U.S. generally

have lower soil moisture than the eastern regions. The IOA and RMSE values in most of the regions are also improved during the three months.

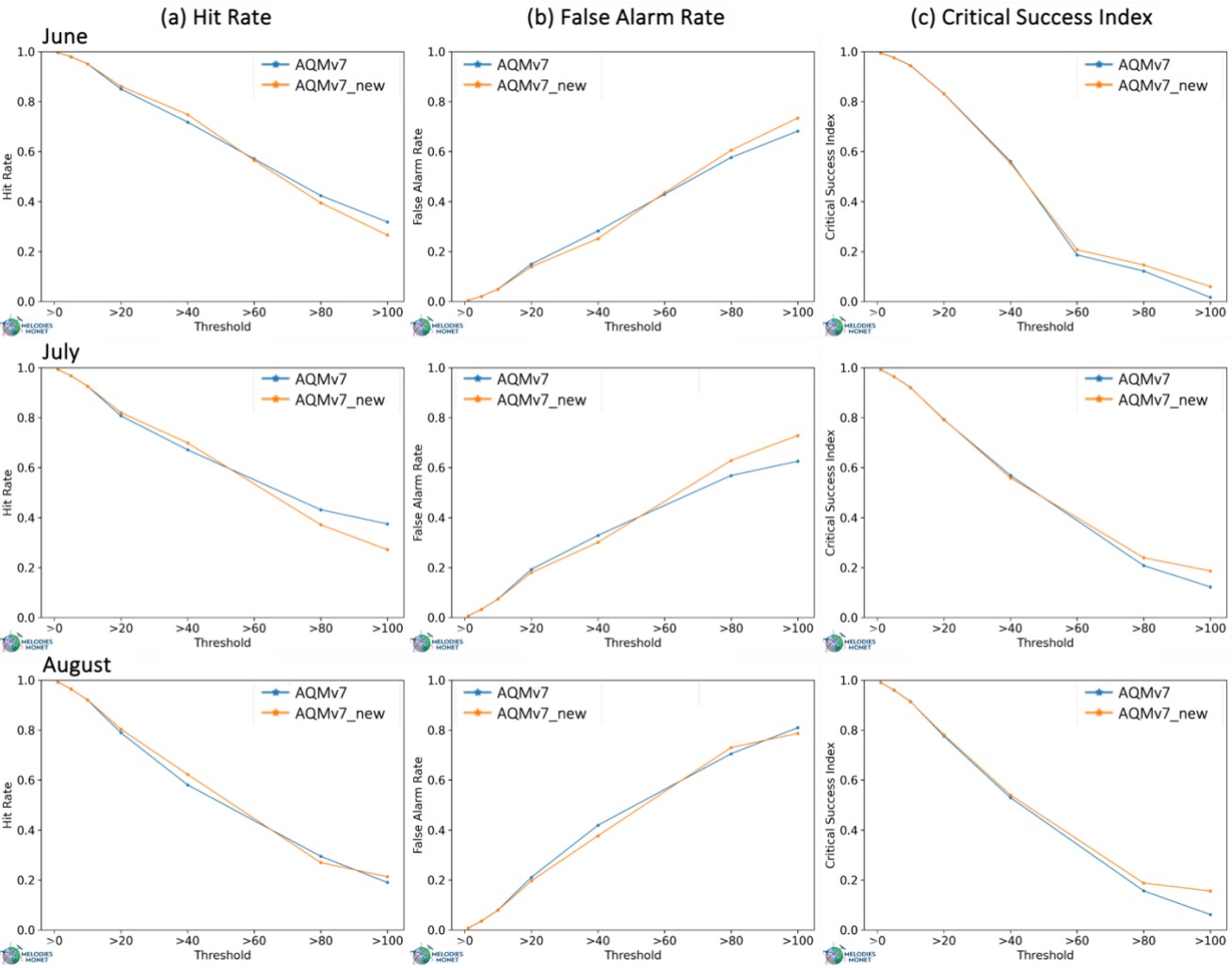

Figure 5: Hit rate (a), false alarm rate (b), and critical success index (c) of hourly O₃ at different thresholds across the CONUS for June (top row), July (middle row) and August (bottom row) 2023.

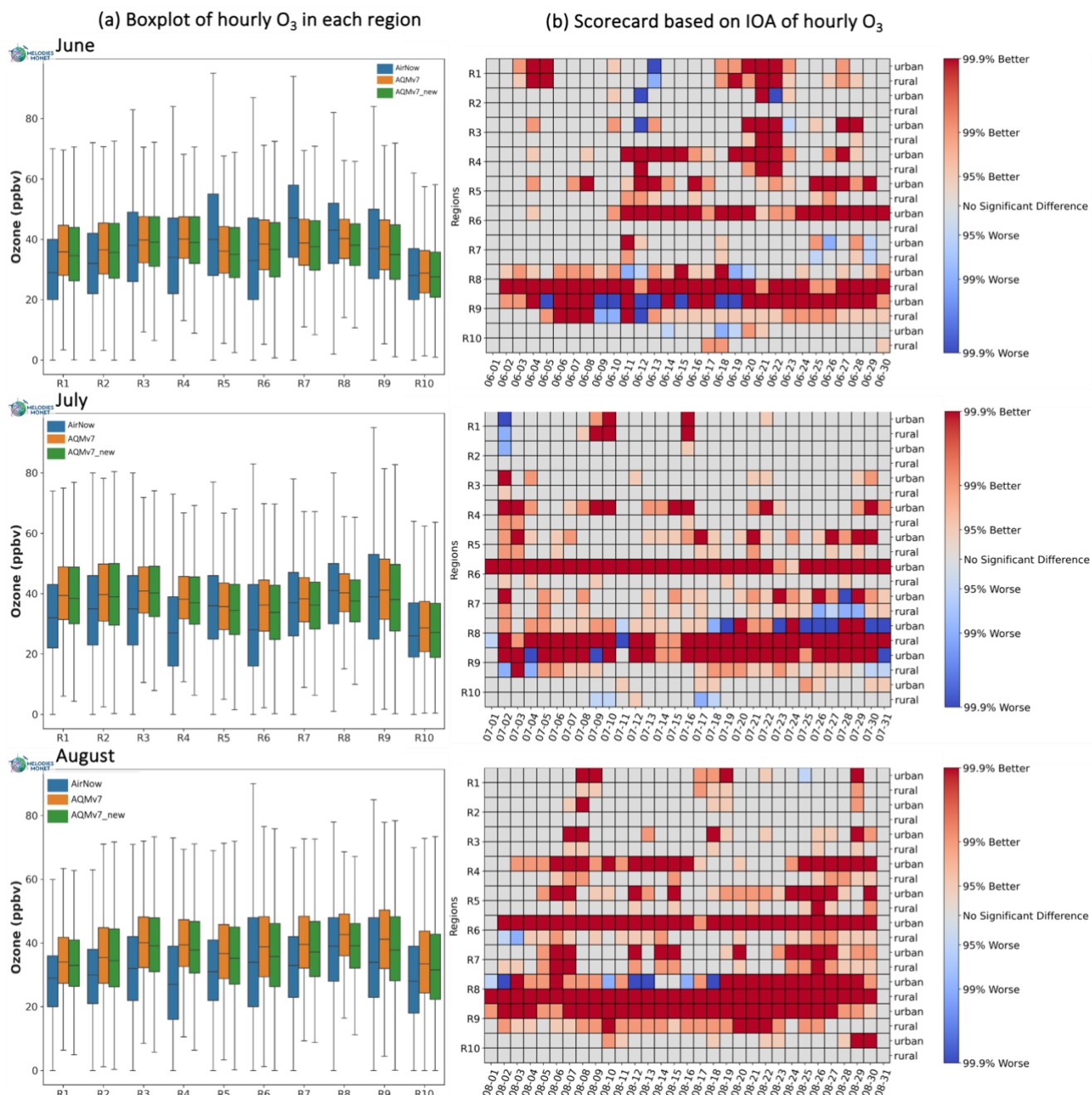

**Figure 6: (a) Boxplot of observed and model-simulated hourly O₃ separated by ten EPA regions. (b) Scorecard plot based on IOA values grouped by urban and rural sites (right axis) within each region (left axis) on each day. Red colors indicate the AQMv7_new model performs better, while blue colors indicate that the AQMv7 model performs better. The saturation of the colors varies by significance levels. Gray color indicate there being no significant difference between two models.**

The regional analysis was also conducted by comparing IOA values between these two models on a daily basis and the results are shown in the scorecard plot (Figure 6b). The IOA is a standardized measure of the degree of model prediction error and is

defined as the ratio of the mean square error to the potential error. The calculation of IOA can be found in the supplementary information. A value of 1 indicates a perfect match between the model and observations, while a value of 0 indicates no agreement at all (Willmott, 1981). The new model has higher IOA values on most of the days from R4 to R9 at a 95% confidence level, although R1-3 and R10 only have improved IOA values on individual days. It is noted that there are some days on which the AQMv7_new model performs worse at both urban and rural sites in a specific region (e.g., June 09 – 10 in

R9). The time series focusing on R9 (Figure S12) reveal that the AQMv7 model generally underestimates $O_3$ on those days and a further reduction in the new model will make the performance worse.

In summary, we compared the model performance of two models in their capability of predicting the spatiotemporal patterns of $O_3$ in the CONUS and found that the updated AQMv7_new model reduces the positive bias and the RMSE values of both hourly and MDA8 $O_3$, indicating an improved model accuracy. The extent of the model performance improvements also differs

by region with the central and southwest areas experiencing the highest enhancement likely due to contributions from both halogen chemistry and dry deposition.

### 4.3 PM$_{2.5}$ evaluation

As shown in Figure 7a, the monthly average of the hourly PM$_{2.5}$ spatial map from AQMv7 displays extremely high values

over eastern and western Canada and the northwestern US due to wildfire emissions. The fire plumes were transported to the northeastern US, especially for the extreme event in June, and led to higher PM$_{2.5}$ levels compared to those in the central and southwestern regions. The negative mean bias of PM$_{2.5}$ at the AirNow sites downstream the wildfire plumes is very high with a value of up to -15 $\mu g/m^3$, where generally in the west-northwest U.S., there are PM$_{2.5}$ overpredictions near fire sources. This result implies that there are substantial uncertainties in wildfire emissions and plume rise, smoke transport, and smoke plume

chemistry for AQMv7. Some sites over the Great Plain and the northeast of August have a relatively smaller positive mean bias of less than 10 $\mu g/m^3$, followed by the close-to-zero mean bias at the sites over the south. The AQMv7_new model also predicts extreme PM$_{2.5}$ values near the wildfire locations and thus shows comparable positive or negative bias as the AQMv7. However, the positive mean bias in the Great Plain and the northeastern area of August is reduced in AQMv7_new, which implies that the overall effect of the model updates is to reduce PM$_{2.5}$ in places with less wildfire impact. Such reductions

inevitably deteriorate the model performance when AQMv7 is unbiased or already underestimates PM$_{2.5}$ at the sites in the southern US.

(a) Monthly mean AQMv7 predicted hourly PM$_{2.5}$ overlaid by AirNow observations (left column) and its difference between predictions and observations at each site (right column)

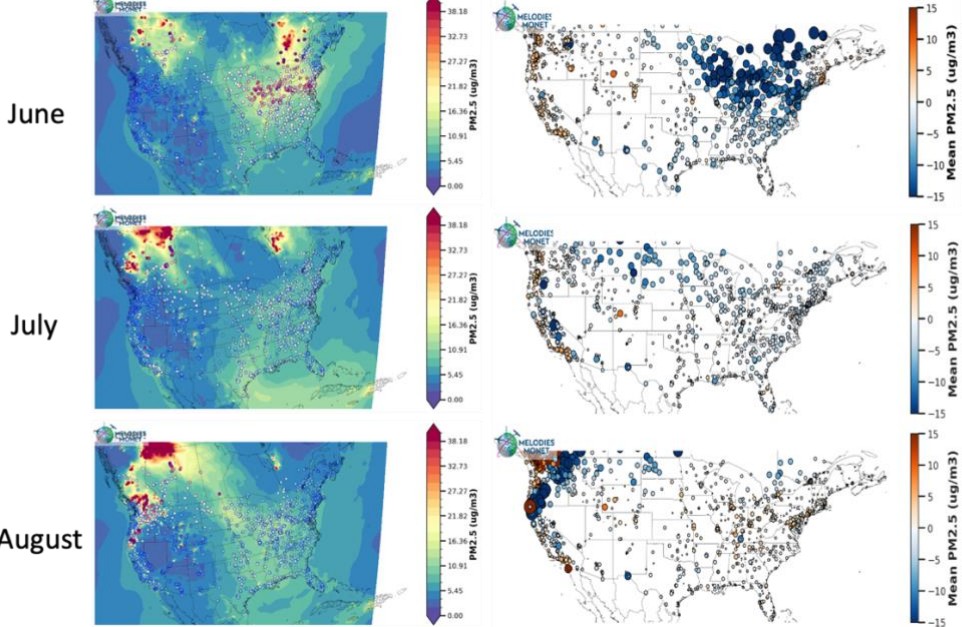

(b) Monthly mean AQMv7_new predicted hourly PM$_{2.5}$ overlaid by AirNow observations (left column) and its difference between predictions and observations at each site (right column)

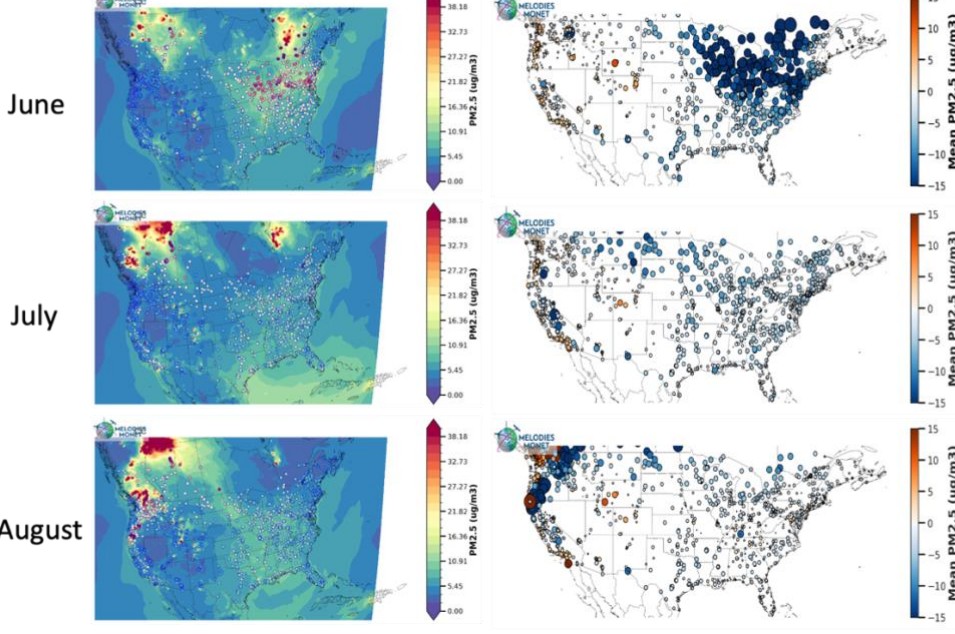

**Figure 7: Same as Figure 3 but for PM$_{2.5}$.**

The hourly and daily time series of the CONUS-mean $PM_{2.5}$ are shown in Figure 8 and their corresponding statistics are summarized in Table 6-8 and Table S4-6. The two models have similar temporal variations and they both miss the high $PM_{2.5}$ episodes in June 6-9, 27-30 and July 4-5, 15-18, 24-27, while better capturing the peak during August 19-21, which are dominated by enhanced fire sources across the U.S. The AQMv7 overall shows an underestimated simulation of hourly $PM_{2.5}$ with a mean bias value of -3.23 $\mu g/m^3$ (-24.13%) for June, -2.06 $\mu g/m^3$ (-19.25%) for July, and -0.61 $\mu g/m^3$ (-5.47%) for

August over the CONUS. The higher negative bias in June and July can be explained by the missed fire-induced high $PM_{2.5}$ values. The AQMv7_new predicts lower $PM_{2.5}$ values at most hours, which increases the mean bias to -4.85 $\mu g/m^3$ (-36.24%), -2.83 $\mu g/m^3$ (-26.39%), and -1.84 $\mu g/m^3$ (-16.42%) from June to August, respectively. Daily $PM_{2.5}$ from the AQMv7_new is also lower on all days, increasing the negative bias from -3.37 $\mu g/m^3$ (-24.74%) to -5.03 $\mu g/m^3$ (-36.98%) for June, from -2.02 $\mu g/m^3$ (-18.96%) to -2.78 $\mu g/m^3$ (-26.07%) for July, and from -0.62 $\mu g/m^3$ (-5.52%) to -1.88 $\mu g/m^3$ (-16.60%) for August.

Similarly, the hourly and daily IOA values are worsened for all three months, with the RMSE value only slightly improved in August.

The hit rate, false alarm rate, and CSI for $PM_{2.5}$ resemble the changes of $O_3$ as the threshold varies from low to high, with a generally decreasing hit rate and CSI and increasing false alarm rate (Figure 9). The AQMv7_new shows a lower hit rate and CSI and a higher false alarm rate in June and July when $PM_{2.5}$ is greater than 60 $\mu g/m^3$, implying an overall worse model

performance. As opposed to June and July, although CSI values in August only slightly increase in AQMv7_new when $PM_{2.5}$ is greater than 40 $\mu g/m^3$, the values of hit rate and false alarm rate become higher and lower compared to AQMv7, respectively, and the changes are bigger at higher thresholds. This indicates that AQMv7_new can better predict August $PM_{2.5}$ at most pollution levels with more improvements found in highly polluted cases. Considering the fire events are better captured in August, the contrasting model performance from our updates between June-July and August highlights the necessity to

improve the representation of wildfire processes for future model developments.

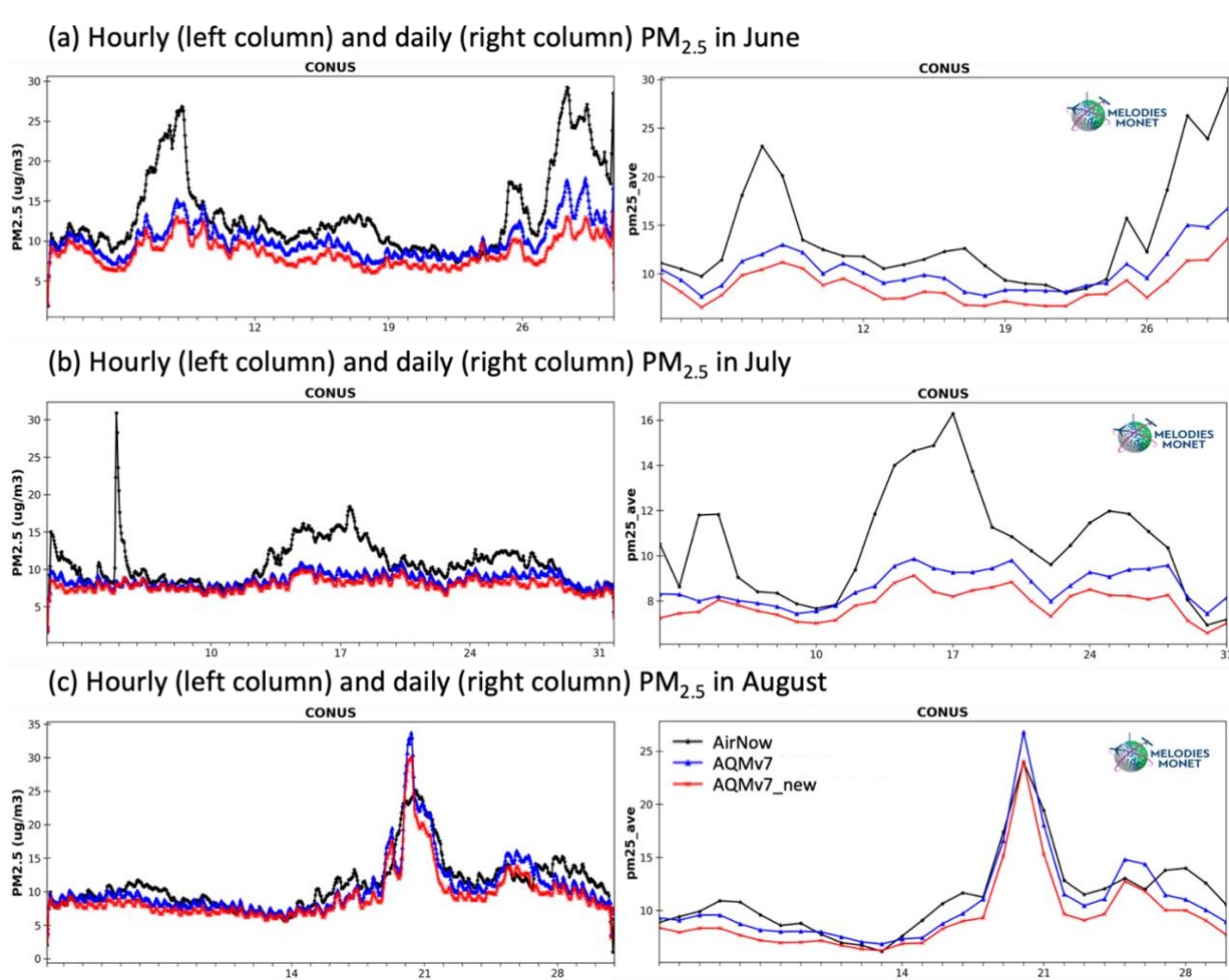

**Figure 8: Same as Figure 4 but for PM₂.₅.**

**Table 6: Hourly PM2.5 evaluation statistics of the AQMv7 and AQMv7_new simulations against the AirNow network in the CONUS and different regions in June 2023. The bold numbers in AQMv7_new indicate an improvement relative to those in AQMv7.**

| Region | Model | MB (µg/m³) | NMB (%) | MdnB (µg/m³) | NMdnB (%) | $R^2$ | RMSE (µg/m³) | IOA |
|--------|-------|-----------|---------|-------------|-----------|-------|-------------|-----|
| CONUS | AQMv7 | -3.23 | -24.13 | -0.04 | -0.47 | 0.32 | 19.78 | 0.60 |
| | AQMv7_new | -4.85 | -36.24 | -0.94 | -12.43 | 0.27 | 20.96 | 0.52 |
| | AQMv7 | 0.65 | 5.82 | 2.38 | 36.70 | 0.29 | 13.95 | 0.66 |

| Region | Model | MB (µg/m³) | NMB (%) | MdnB (µg/m³) | NMdnB (%) | R² | RMSE (µg/m³) | IOA |
|---|---|---|---|---|---|---|---|---|
| Region 1 (northeast) | AQMv7_new | -2.85 | -25.46 | **0.12** | **1.93** | **0.32** | 14.01 | 0.59 |
| Region 2 (NY-NJ) | AQMv7 | -4.78 | -22.23 | 1.55 | 15.48 | 0.43 | 27.21 | 0.68 |
| | AQMv7_new | -8.76 | -40.73 | **-0.93** | **-9.29** | **0.44** | 29.14 | 0.60 |
| Region 3 (mid-Atlantic) | AQMv7 | -10.04 | -39.77 | -2.17 | -15.32 | 0.43 | 31.63 | 0.61 |
| | AQMv7_new | -13.68 | -54.21 | -4.95 | -34.88 | **0.45** | 33.79 | 0.54 |
| Region 4 (southeast) | AQMv7 | -3.92 | -29.38 | -2.32 | -21.66 | 0.34 | 9.12 | 0.58 |
| | AQMv7_new | -5.25 | -39.41 | -3.39 | -31.70 | 0.32 | 9.93 | 0.54 |
| Region 5 (upper Midwest) | AQMv7 | -11.47 | -43.92 | -4.74 | -27.90 | 0.42 | 27.27 | 0.61 |
| | AQMv7_new | -14.94 | -57.23 | -7.36 | -43.28 | 0.33 | 30.62 | 0.49 |
| Region 6 (south) | AQMv7 | -2.90 | -25.79 | -1.84 | -18.62 | 0.14 | 7.34 | 0.54 |
| | AQMv7_new | -3.81 | -33.88 | -2.83 | -28.63 | 0.09 | 8.07 | 0.50 |
| Region 7 (central Great Plain) | AQMv7 | -3.56 | -23.02 | -1.21 | -10.09 | 0.24 | 16.58 | 0.49 |
| | AQMv7_new | -5.86 | -37.88 | -3.22 | -26.82 | 0.19 | 17.78 | 0.40 |
| Region 8 (northern Great Plain) | AQMv7 | -0.4 | -5.75 | -0.05 | -0.93 | 0.02 | 8.98 | 0.34 |
| | AQMv7_new | -0.57 | -8.30 | -0.29 | -5.50 | 0.01 | 11.75 | 0.21 |
| Region 9 (southwest) | AQMv7 | 0.91 | 13.95 | 1.16 | 19.40 | 0.12 | 5.01 | 0.57 |
| | AQMv7_new | **0.64** | **9.71** | **0.96** | **15.94** | 0.12 | **4.91** | 0.57 |
| Region 10 (northwest) | AQMv7 | 2.31 | 54.35 | 1.95 | 54.03 | 0.07 | 5.08 | 0.45 |
| | AQMv7_new | **1.73** | **40.62** | **1.49** | **41.31** | 0.07 | **4.61** | **0.47** |

**Table 7: Same as Table 6 but for July 2023.**

| Region | Model | MB (µg/m³) | NMB (%) | MdnB (µg/m³) | NMdnB (%) | R² | RMSE (µg/m³) | IOA |
|---|---|---|---|---|---|---|---|---|
| CONUS | AQMv7 | -2.06 | -19.25 | -0.20 | -2.65 | 0.21 | 13.95 | 0.52 |
| | AQMv7_new | -2.83 | -26.39 | -0.80 | -10.56 | 0.20 | 14.21 | 0.49 |
| Region 1 (northeast) | AQMv7 | -1.21 | -11.45 | 0.10 | 1.18 | 0.29 | 7.82 | 0.62 |
| | AQMv7_new | -2.77 | -26.17 | -1.18 | -14.40 | 0.22 | 8.55 | 0.53 |
| Region 2 (NY-NJ) | AQMv7 | -1.40 | -11.52 | 0.53 | 5.51 | 0.13 | 12.06 | 0.38 |
| | AQMv7_new | -2.68 | -22.06 | -0.68 | -7.06 | 0.12 | 12.34 | 0.37 |
| Region 3 (mid-Atlantic) | AQMv7 | -1.65 | -14.52 | -0.05 | -0.57 | 0.23 | 8.35 | 0.49 |
| | AQMv7_new | -2.85 | -25.03 | -1.12 | -12.29 | 0.18 | 8.87 | 0.45 |

| Region | Model | MB (µg/m³) | NMB (%) | MdnB (µg/m³) | NMdnB (%) | R² | RMSE (µg/m³) | IOA |
|---|---|---|---|---|---|---|---|---|
| Region 4 (southeast) | AQMv7 | -1.99 | -17.56 | -0.81 | -8.90 | 0.13 | 8.27 | 0.47 |
| | AQMv7_new | -2.84 | -25.04 | -1.48 | -16.29 | **0.15** | 8.41 | **0.48** |
| Region 5 (upper Midwest) | AQMv7 | -2.68 | -21.30 | -0.13 | -1.46 | 0.19 | 11.53 | 0.46 |
| | AQMv7_new | -4.30 | -34.16 | -1.44 | -15.99 | 0.16 | 12.23 | 0.42 |
| Region 6 (south) | AQMv7 | -1.48 | -12.51 | -1.08 | -10.07 | 0.39 | 5.96 | 0.76 |
| | AQMv7_new | -2.63 | -22.29 | -2.14 | -20.02 | 0.31 | 6.64 | 0.69 |
| Region 7 (central Great Plain) | AQMv7 | -2.24 | -21.32 | -0.63 | -7.86 | 0.15 | 9.68 | 0.42 |
| | AQMv7_new | -3.58 | -34.07 | -1.76 | -21.95 | 0.13 | 10.21 | 0.39 |
| Region 8 (northern Great Plain) | AQMv7 | -3.11 | -34.86 | -1.54 | -24.08 | 0.03 | 12.24 | 0.35 |
| | AQMv7_new | **-3.06** | **-34.24** | **-1.45** | **-22.61** | 0.02 | **12.21** | 0.33 |
| Region 9 (southwest) | AQMv7 | -0.31 | -3.67 | 0.28 | 3.98 | 0.07 | 10.55 | 0.40 |
| | AQMv7_new | -0.35 | -4.07 | 0.29 | 4.16 | 0.07 | 10.56 | 0.39 |
| Region 10 (northwest) | AQMv7 | 0.48 | 8.38 | 1.12 | 28.11 | 0.03 | 10.21 | 0.28 |
| | AQMv7_new | **0.33** | **5.80** | **0.95** | **23.63** | **0.04** | **10.15** | 0.28 |

**Table 8: Same as Table 6 but for August 2023.**

| Region | Model | MB (µg/m³) | NMB (%) | MdnB (µg/m³) | NMdnB (%) | R² | RMSE (µg/m³) | IOA |
|---|---|---|---|---|---|---|---|---|
| CONUS | AQMv7 | -0.61 | -5.47 | 0.31 | 4.06 | 0.07 | 32.71 | 0.39 |
| | AQMv7_new | -1.84 | -16.42 | -0.59 | -7.73 | 0.07 | **32.21** | 0.38 |
| Region 1 (northeast) | AQMv7 | 0.90 | 13.35 | 1.22 | 23.50 | 0.24 | 4.13 | 0.68 |
| | AQMv7_new | **-0.83** | **-12.39** | **-0.30** | **-5.83** | 0.14 | 4.37 | 0.59 |
| Region 2 (NY-NJ) | AQMv7 | 1.79 | 21.56 | 1.97 | 27.05 | 0.20 | 5.30 | 0.63 |
| | AQMv7_new | **0.11** | **1.28** | **0.34** | **4.68** | 0.13 | **5.25** | 0.59 |
| Region 3 (mid-Atlantic) | AQMv7 | 1.07 | 11.46 | 1.25 | 15.58 | 0.27 | 4.83 | 0.70 |
| | AQMv7_new | **-0.74** | **-7.89** | **-0.32** | **-3.94** | 0.21 | 4.97 | 0.65 |
| Region 4 (southeast) | AQMv7 | 0.20 | 2.02 | 0.41 | 4.70 | 0.30 | 4.52 | 0.72 |
| | AQMv7_new | -0.88 | -8.97 | -0.55 | -6.33 | 0.26 | 4.71 | 0.69 |
| Region 5 (upper Midwest) | AQMv7 | 0.53 | 4.49 | 0.76 | 7.33 | 0.18 | 8.49 | 0.63 |
| | AQMv7_new | -2.49 | -21.21 | -1.65 | -15.90 | 0.17 | **7.59** | 0.62 |
| Region 6 (south) | AQMv7 | -1.37 | -13.48 | -0.85 | -9.12 | 0.22 | 5.93 | 0.63 |
| | AQMv7_new | -2.19 | -21.61 | -1.80 | -19.34 | 0.20 | 6.39 | 0.62 |

| | | | | | | | | |
|---|---|---|---|---|---|---|---|---|
| Region 7 (central Great Plain) | AQMv7 | 0.95 | 9.48 | 1.12 | 12.34 | 0.12 | 7.13 | 0.56 |
| | AQMv7_new | -1.30 | -12.97 | **-0.98** | **-10.82** | 0.11 | **6.54** | **0.57** |
| Region 8 (northern Great Plain) | AQMv7 | -1.60 | -18.74 | -0.68 | -11.29 | 0.06 | 12.78 | 0.44 |
| | AQMv7_new | -1.82 | -21.31 | -0.82 | -13.64 | 0.04 | 13.58 | 0.38 |
| Region 9 (southwest) | AQMv7 | -0.26 | -3.37 | 0.47 | 7.81 | 0.11 | 10.53 | 0.46 |
| | AQMv7_new | -0.31 | -3.99 | **0.44** | **7.36** | 0.10 | 10.58 | 0.45 |
| Region 10 (northwest) | AQMv7 | -3.10 | -18.65 | 0.31 | 4.90 | 0.11 | 54.54 | 0.47 |
| | AQMv7_new | -3.83 | -23.00 | **0.18** | **2.80** | 0.11 | **51.63** | **0.49** |


The evaluation by each EPA region is illustrated in Figure 10 and the corresponding metrics are listed in Table 6-8. Seven and nine out of the ten regions show underestimated PM$_{2.5}$ values from the AQMv7 simulation for June and July, respectively. The highest negative mean bias is found in the upper Midwest (R5) of June with a value of -11.47 µg/m$^3$ (-43.92%) and the northern Great Plain (R8) of July with a value of -3.11 µg/m$^3$ (-34.86%), suggesting a different fire plume transport pathway mainly from eastern and western Canada for June and July, respectively. For August, the AQMv7 shows a general overestimation in the eastern US (R1-R5) and central Great Plain (R7) with the positive bias values ranging from 0.20 µg/m$^3$ (2.02%) in the southeast (R4) to 1.79 µg/m$^3$ (21.56%) in the New York-New Jersey area (NY-NJ; R2). Regions in the western US (R6, R8-R10) exhibit an overall underestimation of PM$_{2.5}$ with the lowest negative bias of -0.26 µg/m$^3$ (-3.37%) found in the southwest (R9). The highest mean bias of -3.10 µg/m$^3$ (-18.65%) among all the 10 regions lies in the northwest (R10), which can be attributed to wildfires from local sources and southwest Canada. The RMSE value of 54.54 µg/m$^3$ in the southwest is also much higher than those in other regions, which range from 4.13 µg/m$^3$ to 12.78 µg/m$^3$. From the boxplot in Figure 10a, the AQMv7_new predicts a uniformly reduced PM$_{2.5}$ level in all regions, except for R8 of July, which is consistent with the time series in Figure 8. Such effects improve the model performance in regions with a relatively large positive bias, such as R9-10 of June. However, if the overestimation is small (e.g., R1 of June) or if AQMv7 underestimates PM$_{2.5}$, a further reduction in AQMv7_new deteriorates the model performance by increasing the mean bias. Unlike O$_3$, PM$_{2.5}$ experiences a higher level of reduction in the eastern US (R1-R7) for all three months, ranging from 0.82 µg/m$^3$ to 3.98 µg/m$^3$. In contrast, the western US areas (R8-R10) witness a lower reduction by 0.11 µg/m$^3$, 0.12 µg/m$^3$, and 0.49 µg/m$^3$ averaging across three months, respectively. Since the use of AERO7 generally enhances PM$_{2.5}$ mass concentrations (Section 2), such spatial patterns can be explained by the dominating updates to the dry deposition scheme, which increases the deposition velocity of the accumulation mode aerosol by a factor of 10 in forested areas (Pleim et al., 2022), with less enhancement for low-lying vegetation.

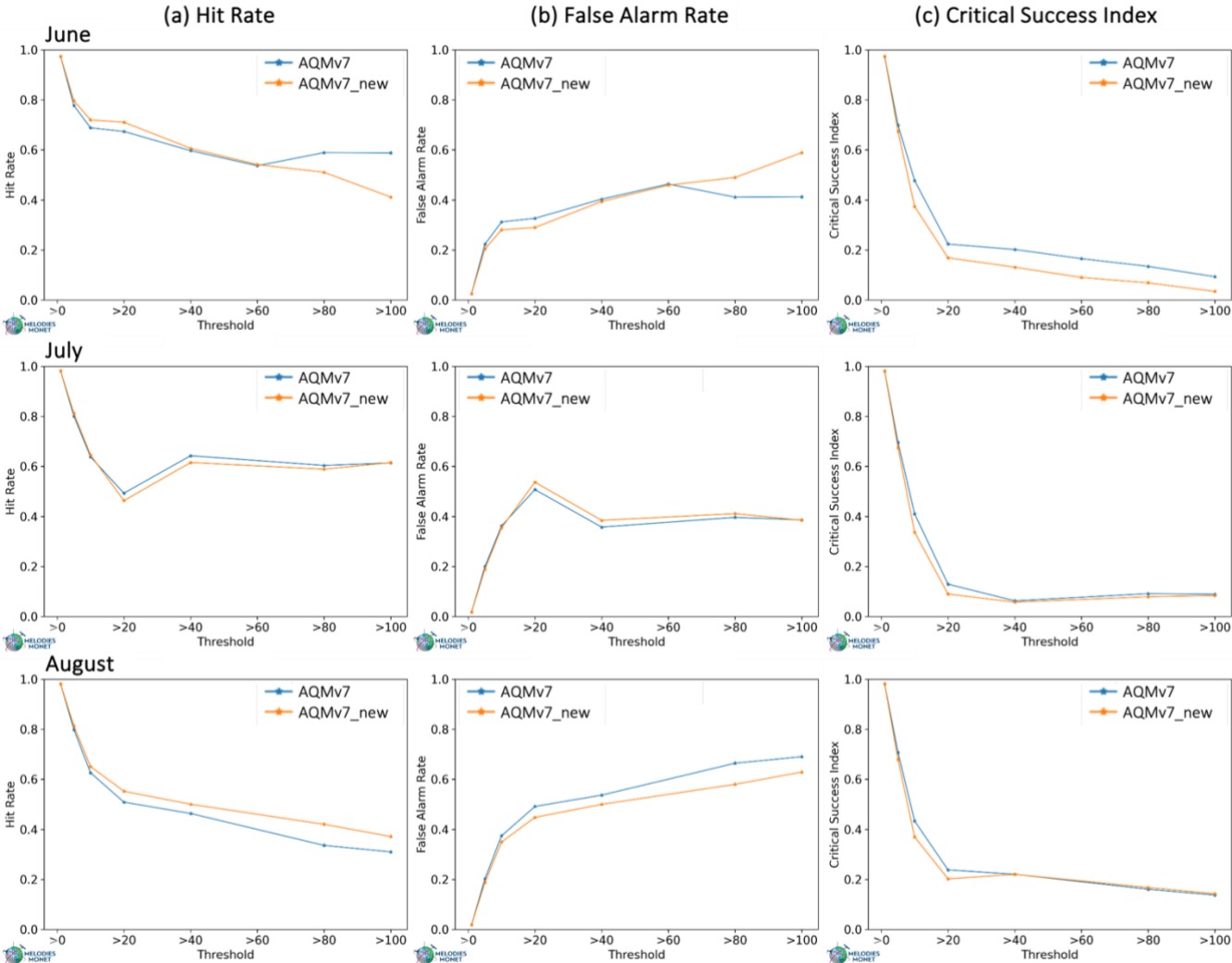

**Figure 9: Same as Figure 5 but for PM$_{2.5}$.**

420

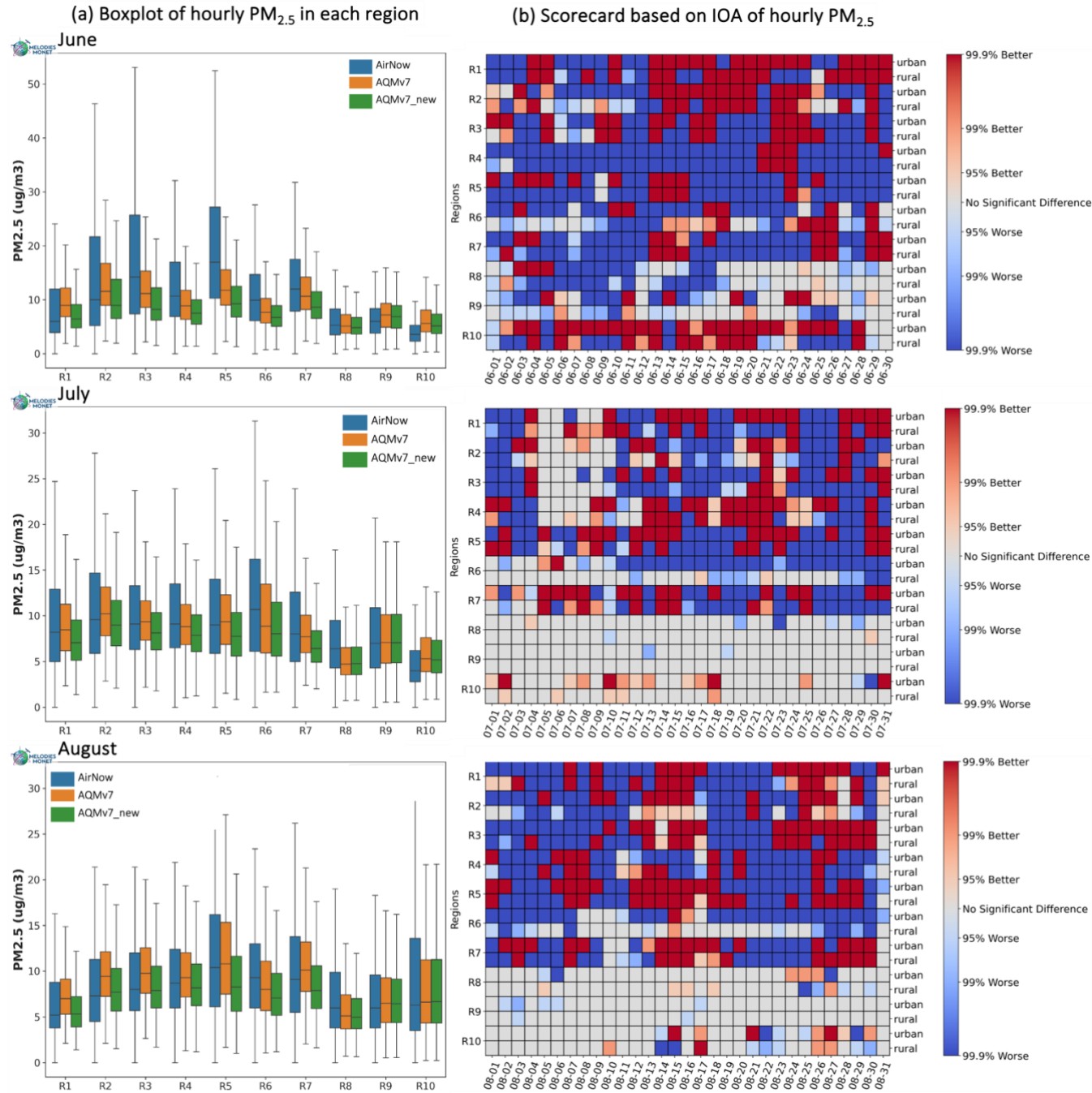

**Figure 10: Same as Figure 6 but for PM$_{2.5}$.**

The scorecard plot in Figure 10b compares the daily values of IOA between the two models in each region. Unlike the considerable differences in the eastern US (R1-R7), positive or negative, most days in the western US (R8-R10) do not have statistically significant changes in July and August. The AQMv7_new seems to only improve IOA on individual days for most of the regions in the three months. This is likely due to the frequent impact from wildfire events, which lead to $PM_{2.5}$ underestimation in the AQMv7 model on the majority of days. Although there is only one to three $PM_{2.5}$ peaks averaged across the CONUS (time series Figure 8) from wildfire plumes, some regions may experience more high $PM_{2.5}$ episodes spanning different dates from the CONUS-mean results and thus make AQMv7 underestimate $PM_{2.5}$ on most days. Here we show an example of R5 for the three months in Figure S13, in which several high $PM_{2.5}$ episodes of greater than 15 µg/m$^3$ are missed by both models. The IOA values are improved out of these episodes, such as July 30-31, in the AQMv7_new model. The underestimation of $PM_{2.5}$ from wildfire can be partly attributed to the fact that gaseous (speciated particulates) fire emissions from the RAVE inventory are scaled from CO ($PM_{2.5}$) and the factors currently used are too small, resulting in lower trace gases and aerosol predictions.

In summary, the AQMv7 demonstrates large bias/error for $PM_{2.5}$ near and downstream of wildfire sources from Canada and the northwestern US, indicating uncertainties in fire emissions and plume rise, transport, and smoke plume chemistry, while there is an overall smaller $PM_{2.5}$ bias in the southern US. The AQMv7_new demonstrates a reduced $PM_{2.5}$ level in all regions, which closes the gap between model and observation in the places where positive biases are found, thus improving the $PM_{2.5}$ predictive accuracy therein. However, the reduction also worsens the model performance in the regions with a negative bias, which becomes more frequent during our study period with the influence from wildfires. The magnitude of the reduction in the AQMv7_new displays an east-to-west discrepancy, which is due to the dependence of the dry deposition velocity on vegetation types introduced by the new scheme.

## 5 Conclusion and discussion

An updated AQMv7 model (AQMv7_new) within the UFS system was developed to incorporate the recent scientific improvements from CMAQv5.4. The evolution of gas and aerosol chemistry in AQMv7_new is primarily influenced by the changes in the CB6 scheme, the introduction of a new aerosol module, and updated air-surface exchange processes. The adoption of CB6r5 in CMAQv5.4 represents an improvement over CB6r3, with updates in halogen chemistry, reaction rates, products, photolysis rates, and the addition of new reactions. The aerosol chemistry scheme, AERO7, introduces key improvements, such as updated monoterpene oxidation yields, organic nitrate formation, water uptake on hydrophilic organic compounds, and a new parameterization for anthropogenic SOA yields. Significant updates in dry deposition processes enhance the representation of air-surface exchange in AQMv7_new. Changes in $O_3$ dry deposition dependence on soil moisture contribute to a more accurate simulation of ambient $O_3$ concentrations. The aerosol dry deposition scheme undergoes

continuous refinement, incorporating factors like leaf area index (LAI) and impaction efficiency based on land use categories. Structural changes in the IO framework of CMAQ, such as the DESID and CIO modules contribute to an improved computational efficiency and the ease of maintenance. The ELMO module in CMAQv5.4 further streamlines the synthesis of model output parameters, reducing the need for post-processing tools.

To test the performance of the AQMv7_new, a three-month simulation in June-August 2023 was conducted over North America and an air quality evaluation was performed for the CONUS in comparison to the surface $O_3$ and $PM_{2.5}$ observations at AirNow sites. AQMv7_new demonstrates improved simulation of $O_3$ concentrations, reflecting a CONUS-mean better spatiotemporal agreement with observations. Generally, there is a nationwide decrease in $O_3$ mixing ratios, mainly reducing the persistent high positive bias observed at coastal sites for AQMv7. Temporally, the AQMv7_new addresses the persistent positive bias in peak values at noon and low values at night, leading to a reduction of 8%-12% in the overprediction of MDA8 $O_3$. While AQMv7 with CMAQ 5.2.1 chemistry tends to overestimate hourly $O_3$ concentrations in the EPA regions with small or no wildfire influence, AQMv7_new with CMAQ 5.4 exhibits a universal shift in the statistical distribution to the lower end, thus reducing the positive bias across these regions. The central and southwestern regions particularly benefit from the model updates, possibly due to the enhanced $O_3$ dry deposition velocity over dry soil and the increased halogen-mediated $O_3$ loss over the sea.

The spatial distribution of monthly average $PM_{2.5}$ concentrations reflects extreme values over the eastern and western Canada and the northwestern US, attributed to wildfire emissions, which introduces substantial uncertainties in the model as indicated by the high mean bias values at the AirNow sites close to and downwind wildfire sources. AQMv7_new generally predicts lower $PM_{2.5}$ values averaged across the CONUS domain, which reduces the positive bias in the northeast for August. Improvements are also found in August for hit rate and false alarm rate at high thresholds, suggesting a better predictive accuracy of $PM_{2.5}$, particularly in highly polluted scenarios when wildfire events are captured by the model. By contrast, the generally worse performance in June and July is likely a result of missing the full extent of fire events in both models. The region-specific evaluation highlights a general underestimation over most of the areas in June and July while an overestimation in the eastern US and an underestimation in the western US by AQMv7 in August, with the AQMv7_new uniformly reducing $PM_{2.5}$ levels across all regions. This reduction improves the predictive accuracy in regions with positive bias but exacerbates the negative bias in regions where AQMv7 already underestimated $PM_{2.5}$. Furthermore, the magnitude of the reduction displays an east-to-west discrepancy: higher reduction in the east and lower in the west. This spatial pattern can be attributed to the changes in the dry deposition scheme, which greatly increases the dry deposition rate over forests for the accumulation mode aerosol.

The NMB of AQMv7_new simulated MDA8 $O_3$ and daily $PM_{2.5}$ over the CONUS is -4.28%, 3.89%, 10.75% and -36.98%, -26.07%, -16.60% from June to August, respectively. Except for the high negative $PM_{2.5}$ bias in June, these values fall in the

benchmark criteria of ±15% for MDA8 $O_3$ and ±30% for daily $PM_{25}$ as suggested by Emery et al. (2017) by summarizing the model performance statistics reported from 2005 to 2015 in the CONUS. This highlights the challenges and uncertainties persisting in accurately capturing the complex dynamics of wildfire emissions and their influence on air quality. The AQMv7_new model cannot improve upon the exacerbated $PM_{2.5}$ predictions near and downstream of wildfire sources (e.g., northeast and west-northwest U.S.), partly due to its current small factors to scale fire-related emission species from CO and total $PM_{2.5}$ in the simulations. Continuous efforts should be made to reduce the uncertainties of wildfire emissions and test cases can be conducted to adjust the RAVE emission factors based on more intensive field campaigns and measurements within smoke plumes. The current UFS-AQM system has limited capabilities in diagnostics and can only write out species concentrations and AOD. This limits our current study to only a qualitative inference that the performance changes are driven by lumped updates to the chemistry, and/or dry deposition schemes based on the CMAQ release notes. However, the verification results in this study showed that the changes from AQM_v7 to AQMv7_new behave similarly to that on the WRF-CMAQ: version 5.2.1 versus 5.4. More process-related diagnostics and tools are currently being added to UFS-AQM to better interpret the performance changes by quantitively attributing them to various processes, such as chemical productions and destructions, dry deposition, and transport. In addition, longer simulations covering multiple seasons and a more comprehensive evaluation with different observational platforms (e.g., surface sites, ozonesondes, aircraft, lidar, and satellite) are also ongoing for a more thorough investigation of the AQM and impacts of the model updates described here. Further refinements to the coupled CCPP physics (e.g., GFS) and the critical driving meteorological parameters are needed, which inherently interact with natural emissions in addition to wildfire, such as biogenic VOCs, soil NO, windblown dust, oceanic dimethyl sulfide (DMS), and lightning NOx emissions, are also highly needed. This study shows that the UFS-AQM framework can well accommodate the community air quality model, like CMAQ, as well as its latest upgrade. The results of this upgrade are consistent with those shown in the WRF-CMAQ systems. This method is proven to be viable for ESMF coupling for different dynamics, physics and chemistry with a hierarchal coding infrastructure linked across authorized repositories at collaborating agencies. Although we did not include some functions of the original CMAQ, such as the decoupled direct method in three dimensions (DDM-3D) (Zhang et al., 2012), Integrated Source Apportionment Method (ISAM) (Kwok et al., 2015) in the UFS-AQM model due to its novel ESMF coupling , the online CMAQ prediction model within this framework yields overall reasonable results. As the UFS-AQM model is current operational air quality forecast system of NOAA, this study underscores the importance of ongoing scientific investigations, refinement, and quality assurance processes in atmospheric modelling to ensure reliable predictions and advance our understanding of the intricate interactions driving air quality variability.

**Code and data availability**

The UFS-AQMv7 source codes are available on the following GitHub repository GitHub - ufs-community/ufs-srweather-app at production/AQM.v7 (last access: 15 March 2024). The AQMv7_new codes are reposited at https://zenodo.org/records/10833128 (last access: 19 March 2024) and can also be downloaded via a GitHub tag GitHub - noaa-oar-arl/AQM at CMAQ54_Paper (last access: 15 March 2024).

**Author contributions**

WL conducted the model updates and drafted the initial manuscript. BT contributed to model updates, conducted the three months model runs, and performed model evaluation for gas and aerosol species. PCC guided BT for model runs and conducted meteorology evaluation. PCC, YT, BB, ZM, KW, JH, and RM contributed to model updates, project methodology, analyses, and evaluation. BB, DT, IS, PCC, and YT contributed to project administration, funding acquisition, and supervision. All authors contributed to the interpretation of the results and revisions of the paper.

**Competing interest**

The contact author has declared that neither they nor their co-authors have any competing interests.

**Financial support**

This study was co-funded by NOAA grants NA24NESX432C0001 and NA19NES4320002 (Cooperative Institute for Satellite Earth System Studies - CISESS) at the University of Maryland/ESSIC and NOAA IIJA (NA22NES4050024I/121327-Z7648201), DSRA (NA22NES4050023D/119812-Z7646201). We acknowledge the developers for creating the useful evaluation tools of MELODIES-MONET and AMET.

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
