# Peer review of "Updates and evaluation of NOAA's online-coupled air quality model version 7 (AQMv7) within the Unified Forecast System"

_Geoscientific Model Development, 2024_

## Author Comment (AC1)

**Reply to Reviewers**

We sincerely appreciate all the reviewers for their constructive comments to improve the manuscript. Their comments are reproduced below followed by our responses in blue. The corresponding edits in the manuscript are highlighted with track changes.

**Reviewer #1:**

General Comments:

The manuscript "Updates and evaluation of NOAA's online-coupled air quality model version 7 (AQMv7) within the Unified Forecast System" provides a thorough evaluation of the updated NOAA's air quality model version 7 (AQMv7) within the Unified Forecast System (UFS), incorporating the recent scientific improvements from the CMAQv5.4. The authors conducted two experiments in August 2023 to assess the performance of updates. The updated version (AQMv7_new) significantly improved the spatial and temporal persistent high positive biases of the ozone mixing ratios. The AQMv7_new demonstrated reduced PM2.5 in all regions of Contiguous United States (CONUS). This study indicates that the community air quality model of CMAQ could be well accommodated in the UFS-AQM framework. This manuscript is well written and offers a valuable contribution into the evolution of NOAA's air quality models. Therefore, it is recommended to accept this manuscript after addressing some minor revisions.

Minor Revisions:

(1). Is there aerosol/chemistry feedback to the host atmospheric model in AQMv7 and AQMv7_new?

**Response**: No. It is currently a one-way model to use UFS to drive aerosol and chemistry only. We have clarified this in the revised manuscript (Lines 107).

(2). The wet deposition is also very important for the aerosol and chemical processes. However, the wet deposition is not mentioned in this manuscript. What kind of wet deposition is used in AQMv7_new?

**Response**: The in-cloud chemistry and wet scavenging are handled in CMAQ based on Chang et al. (1987), while the convective removal through precipitation is processed in the Common Community Physics Package (CCPP) physics. The information has now been added in Lines 105-106.

(3). Lines 104-105: "while other transport terms, such as advection and diffusion, are more appropriately handed in the FV3 physics". Please clarify whether the advection and diffusion are handled in the FV3 physics or dynamics?

**Response**: Advection is performed by the FV3 dynamical core. Mixing/diffusion is performed by the PBL physics scheme within CCPP. We clarified it in the revised manuscript.

(4). Line 409: replace "AQMv_new" with "AQMv7_new" for consistency.

**Response**: Thanks for catching the typo. It is corrected now.

**Reviewer #2**

General Comments:

The paper reports on an update of the CMAQ model with respect to chemistry and dry deposition. CMAQ is used as column model as part of UFS air quality forecasting system. The model changes are described and the performance of the UFS air quality forecast with and without the CMAQ update are evaluated with surface ozone and PM2.5 concentrations over the CONUS for August 2023. The evaluation shows reduced ozone and PM2.5 concentrations with the upgrade, which means an improved performance for ozone and more mixed results for PM2.5.

The paper is clear and well written and gives a good overview of the model updates. A considerable limitation of the the paper is the evaluation period (August 2023) of just one month in summer. At least a winter month needs to be included to fully understand the impact of the changes for PM2.5. After all, the UFS air quality forecasting system is an operational system that probably runs continuously over the whole year. Also, some of the discussed changes to ozone deposition over snow can only be evaluated in a winter period. I also believe that one month is too short to derive robust statistics. So it is recommended as good scientific practice to do the evaluation for a 3-months summer and winter period. It is also worth considering that the chosen month was affected by wildfire emissions but none of the upgraded elements seemed to have the potential to improve the AQ forecast in a wildfire dominated situation.

Response: The reviewer's points are well taken. We focused on summer months first because both $O_3$ and $PM_{2.5}$ exceedance can occur during warm seasons. An evaluation during winter months is a good supplement to fully understand our model updates and we will leave it for future work. The statements about ozone deposition over snow have been removed to avoid confusion. We agree that one summer month may not be long enough for a thorough evaluation. Therefore, we expanded the simulation to three months from June to August of 2023 and evaluated the model updates for each month. We also fixed a bug causing the missing of some point source emissions in our updates. All the major figures and statistics have been updated in the revised manuscript. The main conclusion remains the same with improved performance for $O_3$ simulation and mixed results for $PM_{2.5}$ due to frequent influences from wildfires. Although the updates do not improve the air quality simulation when wildfire occurs, our evaluation highlights an underestimation of both $O_3$ and $PM_{2.5}$ downwind fire emission sources and provides valuable insight into future model development.

Specific Comments:

(1) L 124: Please provide more details on this aspect. Is this a deposition term or a gas-phase chemistry term?

Response: The simple halogen chemistry, expressed as a first-order ozone loss to sea water is a chemistry term. Although only detailed halogen chemistry involves gas-phase and heterogeneous reactions, the simplified halogen chemistry is developed using the annual hemispheric CMAQ results obtained with and without detailed halogen chemistry. The revised simple halogen-mediated first-order rate constant for ozone loss (units=$s^{-1}$) is a function of atmospheric pressure P (units=atm) is $K_{O3}$ (P) = min (2.6750 $\times$ $10^{-06}$, 2.8964 $\times$ $10^{-11}$ e $^{11.9978 \times P}$). The description was added in Line 127 in the revised manuscript.

(2) L 143: Is the increase in organic aerosol and PM2.5 primarily confirmed by the evaluation with PM2.5 observations?

**Response**: The AERO7 replaces the Odum 2-product monoterpene SOA in AERO6 with updated yields based on more recent experiments by Saha and Greishop (2016). The new yields are represented using a volatility basis set (VBS) fit and applied to both OH and ozone oxidation of monoterpenes, which allows for prompt formation of low-volatility material and is more consistent with recent observations. The increase in organic aerosol and PM2.5 mass was confirmed by comparing the speciated $PM_{2.5}$ with both the old CMAQ v5.2.1 and AQS observations (Appel et al., 2021).

(3) L 151: The impact of this change can only be demonstrated with a wintertime simulation.
**Response**: We removed the texts about ozone deposition changes over snow to avoid confusion since our evaluation is for summer months only.

(4) L 160: Please provide here or in table 1 information about the used land cover and vegetation data.
**Response**: Thanks for the suggestion. The land cover is based on the 20-category IGBP (International Geosphere-Biosphere Program) classification mechanism from Noah land surface model (LSM), which also provides vegetation parameters, such as LAI. The information is updated in Table 1.

(5). L 216: Please provide more details on the calculation of the IOA.
**Response**: The formula for IOA calculation is:

$$IOA = 1 - \left[ \frac{\sum_{1}^{n} (O - M)^2}{\sum_{1}^{n} \left( \left| M - \overline{O} \right| + \left| O - \overline{O} \right| \right)^2} \right]$$

where M and O represent the predicted and observed concentrations, respectively. This formula was added to the supplementary file.

(6). L 235: Including an evaluation of the meteorology is an interesting aspect and I would recommend to move some plots from the appendix to the main paper. Please confirm that the error of the meteorological fields is exactly the same for both the reference and the upgrade. Please add more details what ozone and PM2.5 error are mostly likely to be introduced by the increased stability. Is there a possibility that the CMAQ upgrades compensates error coming from the meteorology?
**Response**: Thanks for the suggestion. We moved the surface meteorology evaluation plot and statistics table for August to the main text as a new Section 4.1. The June and July meteorology was also assessed and added to the supplementary file. The results are similar to those of August. We kept the same GFS version for the two simulations and confirmed the meteorological fields are identical based on our ongoing analysis comparing with field campaign observations. There is no feedback from aerosol/chemistry to the host atmospheric model, so the performance changes from our model updates do not arise from meteorological differences.

During stable conditions, the vertical mixing of air is limited, trapping pollutants close to the ground and allowing them to accumulate. This can lead to higher ground-level $O_3$ and $PM_{2.5}$

concentrations, especially in urban areas (He et al., 2017; Deak et al., 2021). We added these explanations to Line 239.

(7). L 305. Please discuss in more detail the contributions from deposition and/or halogen chemistry. I got the impression halogen chemistry is more important in coastal areas.
**Response**: In our new three-month simulation, ozone has a higher sensitivity to the model updates in the central and southwest regions (R6-R9), of which R6 (such as Texas and Luisiana) and R9 (such as California) contain large coastal areas. The halogen chemistry updates reduce ozone over sea water, which can be transported into the central U.S. dominated by southerly winds in summer, such as the Great Plain low-level jet (Zhu and Liang, 2013). In addition, the added dependence of ozone dry deposition velocity to soil moisture leads to more ozone uptake by dry soil than wet soil (Appel et al., 2021) and the central and western U.S. generally have lower soil moisture than the eastern regions We briefly described it in Line 317-322 in the revised manuscript.

(8). L 433: Please mention earlier that the fire emissions do not include an VOC, or even not other gases?
**Response**: The VOC emission from fire is turned on in our updated simulations for both AQMv7 and AQMv7_new and the analyses in the revised manuscript. The RAVE VOC emissions are scaled from CO and our evaluation indicates that these scaling factors currently used are too small. As mentioned in the discussion section, we plan to test different factors in the future to improve the model performance under the impact of wildfire. Other gas emissions, such as $NO_x$ and $SO_2$, are also used in both simulations. We mentioned this change earlier in Line 200 in the revised manuscript.

**References:**
Appel, K. W., Bash, J. O., Fahey, K. M., Foley, K. M., Gilliam, R. C., Hogrefe, C., Hutzell, W. T., Kang, D., Mathur, R., Murphy, B. N., Napelenok, S. L., Nolte, C. G., Pleim, J. E., Pouliot, G. A., Pye, H. O. T., Ran, L., Roselle, S. J., Sarwar, G., Schwede, D. B., Sidi, F. I., Spero, T. L., and Wong, D. C.: The Community Multiscale Air Quality (CMAQ) model versions 5.3 and 5.3.1: system updates and evaluation, Geosci. Model Dev., 14, 2867–2897, https://doi.org/10.5194/gmd-14-2867-2021, 2021.

Chang, J. S., Brost, R. A., Isaksen, I. S. A., Madronich, S., Middleton, P., Stockwell, W. R., & Walcek, C. J. (1987). A three-dimensional Eulerian acid deposition model: Physical concepts and formulation. Journal of Geophysical Research: Atmospheres, 92(D12), 14681-14700.

Deak, G., Raischi, N., Lumînăroiu, L., Raischi, M., Vladut, N. V., Prangate, R., & Noor, N. M. (2022, January). The Influence of Atmospheric Stability on PM10 Concentrations in Urban Areas. In Proceedings of the 3rd International Conference on Green Environmental Engineering and Technology: IConGEET 2021, Penang, Malaysia (pp. 51-55). Singapore: Springer Nature Singapore.

He, J., Gong, S., Yu, Y., Yu, L., Wu, L., Mao, H., ... & Li, R. (2017). Air pollution characteristics and their relation to meteorological conditions during 2014–2015 in major Chinese cities. Environmental pollution, 223, 484-496.

Saha, P. K. and Grieshop, A. P.: Exploring Divergent Volatility Properties from Yield and Thermodenuder Measurements of Secondary Organic Aerosol from α-Pinene Ozonolysis, Environmental Science & Technology, 50, 5740-5749, https://doi.org/10.1021/acs.est.6b00303, 2016.

Zhu, J. and Liang, X.-Z.: Impacts of the Bermuda High on Regional Climate and Ozone over the United States, Journal of Climate, 26, 1018–1032, https://doi.org/10.1175/JCLI-D-12-00168.1, 2013.